# Towards conversational diagnostic artificial intelligence

Tao Tu[1,3 ✉], Mike Schaekermann[1,3 ✉], Anil Palepu[1,3], Khaled Saab[1], Jan Freyberg[1], Ryutaro Tanno[2], Amy Wang[1], Brenna Li[1], Mohamed Amin[1], Yong Cheng[2], Elahe Vedadi[1], Nenad Tomasev[2], Shekoofeh Azizi[2], Karan Singhal[1], Le Hou[1], Albert Webson[2], Kavita Kulkarni[1], S. Sara Mahdavi[2], Christopher Semturs[1], Juraj Gottweis[1], Joelle Barral[2], Katherine Chou[1], Greg S. Corrado[1], Yossi Matias[1], Alan Karthikesalingam[1,4 ✉] & Vivek Natarajan[1,4 ✉]

At the heart of medicine lies physician–patient dialogue, where skilful history-taking enables effective diagnosis, management and enduring trust[1,2]. Artificial intelligence (AI) systems capable of diagnostic dialogue could increase accessibility and quality of care. However, approximating clinicians' expertise is an outstanding challenge. Here we introduce AMIE (Articulate Medical Intelligence Explorer), a large language model (LLM)-based AI system optimized for diagnostic dialogue. AMIE uses a self-play-based[3] simulated environment with automated feedback for scaling learning across disease conditions, specialties and contexts. We designed a framework for evaluating clinically meaningful axes of performance, including history-taking, diagnostic accuracy, management, communication skills and empathy. We compared AMIE's performance to that of primary care physicians in a randomized, double-blind crossover study of text-based consultations with validated patient-actors similar to objective structured clinical examination[4,5]. The study included 159 case scenarios from providers in Canada, the United Kingdom and India, 20 primary care physicians compared to AMIE, and evaluations by specialist physicians and patient-actors. AMIE demonstrated greater diagnostic accuracy and superior performance on 30 out of 32 axes according to the specialist physicians and 25 out of 26 axes according to the patient-actors. Our research has several limitations and should be interpreted with caution. Clinicians used synchronous text chat, which permits large-scale LLM–patient interactions, but this is unfamiliar in clinical practice. While further research is required before AMIE could be translated to real-world settings, the results represent a milestone towards conversational diagnostic AI.

The dialogue between the physician and the patient is fundamental to effective and compassionate care. The medical interview has been termed "the most powerful, sensitive, and most versatile instrument available to the physician"[2]. In some settings, it is believed that 60–80% of diagnoses are made through clinical history-taking alone[6]. The physician–patient dialogue extends beyond history-taking and diagnosis—it is a complex interaction that establishes rapport and trust, serves as a tool for addressing health needs and can empower patients to make informed decisions that account for their preferences, expectations and concerns[7]. While there is wide variation in communication skills among clinicians, well-trained professionals can wield considerable skills in clinical history-taking and the wider 'diagnostic dialogue'. However, access to this expertise remains episodic and globally scarce[8].

Recent progress in general-purpose large language models (LLMs)[9–11] has shown that artificial intelligence (AI) systems have the capability to plan, reason and incorporate relevant context enough to hold naturalistic conversations. This progress affords an opportunity to rethink the possibilities of AI in medicine towards the development of fully interactive conversational AI. Such medical AI systems would understand clinical language, intelligently acquire information under uncertainty and engage in natural, diagnostically useful medical conversations with patients and those who care for them. The potential real-world utility of AI systems capable of clinical and diagnostic dialogue is broad, with the development of such capabilities possibly improving access to diagnostic and prognostic expertise, thus improving the quality, consistency, availability and affordability of care. A health equity-centric approach to integrating such technology into existing workflows, which implies work in the development, implementation and policy stages, may have the potential to help realize better health outcomes (particularly for populations facing healthcare disparities).

However, while LLMs have been shown to encode clinical knowledge and have proven capable of highly accurate single-turn medical question-answering[12–14], their conversational capabilities have been tailored to domains outside clinical medicine[15,16]. Earlier work in LLMs for

[1]Google Research, Mountain View, CA, USA. [2]Google DeepMind, Mountain View, CA, USA. [3]These authors contributed equally: Tao Tu, Mike Schaekermann, Anil Palepu. [4]These authors jointly supervised this work: Alan Karthikesalingam, Vivek Natarajan. ✉e-mail: taotu@google.com; mikeshake@google.com; alankarthi@google.com; natviv@google.com

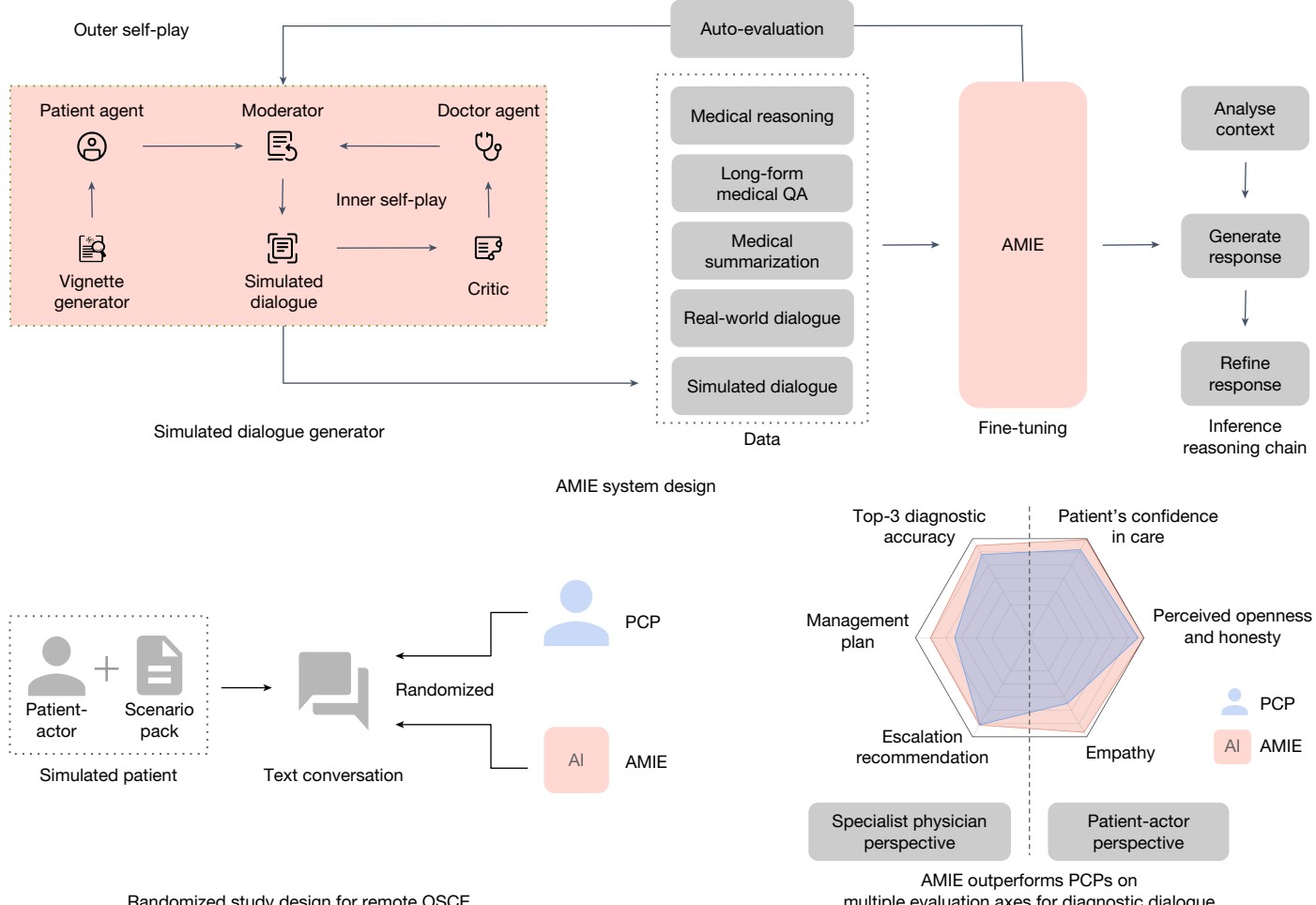

**Fig. 1 | Overview of contributions.** AMIE is a conversational medical AI optimized for diagnostic dialogue. It is instruction fine-tuned with a combination of real-world and simulated medical dialogues, alongside a diverse set of medical reasoning, question-answering (QA) and summarization datasets. Notably, we designed a self-play-based simulated dialogue environment with automated feedback mechanisms to scale AMIE's capabilities across various medical contexts and specialties. Specifically, this iterative self-improvement process consisted of two self-play loops: (1) an 'inner' self-play loop, where AMIE leveraged in-context critic feedback to refine its behaviour on simulated conversations with an AI patient agent; and (2) an 'outer' self-play loop where

the set of refined simulated dialogues were incorporated into subsequent fine-tuning iterations. During online inference, AMIE used a chain-of-reasoning strategy to progressively refine its response, conditioned on the current conversation, to arrive at an accurate and grounded reply to the patient in each dialogue turn. We designed and conducted a blinded remote OSCE with validated patient-actors interacting with AMIE or PCPs by means of a text chat interface. Across multiple axes, corresponding to both specialist physician (30 out of 32) and patient-actor (25 out of 26) perspectives, AMIE was rated as superior to PCPs while being non-inferior on the rest.

health[12–14,17,18] has not yet rigorously examined the clinical history-taking and diagnostic dialogue capabilities of AI systems or contextualized this by comparison to the extensive capabilities of practicing generalist physicians.

Clinical history-taking and diagnostic dialogue, through which clinicians derive diagnosis and management plans, represent a complex skill[1] whose optimal conduct is highly dependent on context. Thus, multiple evaluation axes are needed to assess the quality of a diagnostic dialogue, including the structure and completeness of the elicited history, diagnostic accuracy, the appropriateness of management plans and their rationale, and patient-centred considerations, such as relationship-building, respect for the individual and communication efficacy[19]. If the conversational potential of LLMs is to be realized in medicine, there is an important unmet need to better optimize the development and evaluation of medical AI systems for characteristics such as these, which are unique to history-taking and diagnostic dialogue between clinicians and patients.

Here we detail our progress towards a conversational medical AI system for clinical history-taking, diagnostic reasoning and communication efficacy. We also outline some key limitations and directions for future research.

Our key contributions (Fig. 1) are summarized here. We first introduced AMIE (Articulate Medical Intelligence Explorer), an LLM-based AI system optimized for clinical history-taking and diagnostic dialogue. To scale AMIE across a multitude of specialties and scenarios, we developed a self-play-based simulated diagnostic dialogue environment with automated feedback mechanisms to enrich and accelerate its learning process. We also introduced an inference time chain-of-reasoning strategy to improve AMIE's diagnostic accuracy and conversation quality. Then we developed a pilot evaluation rubric to assess the history-taking, diagnostic reasoning, communication skills and empathy of diagnostic conversational medical AI, encompassing both clinician-centred and patient-centred metrics. Next we designed and conducted a blinded, remote objective structured clinical examination (OSCE) study (Fig. 2) using 159 case scenarios from clinical providers

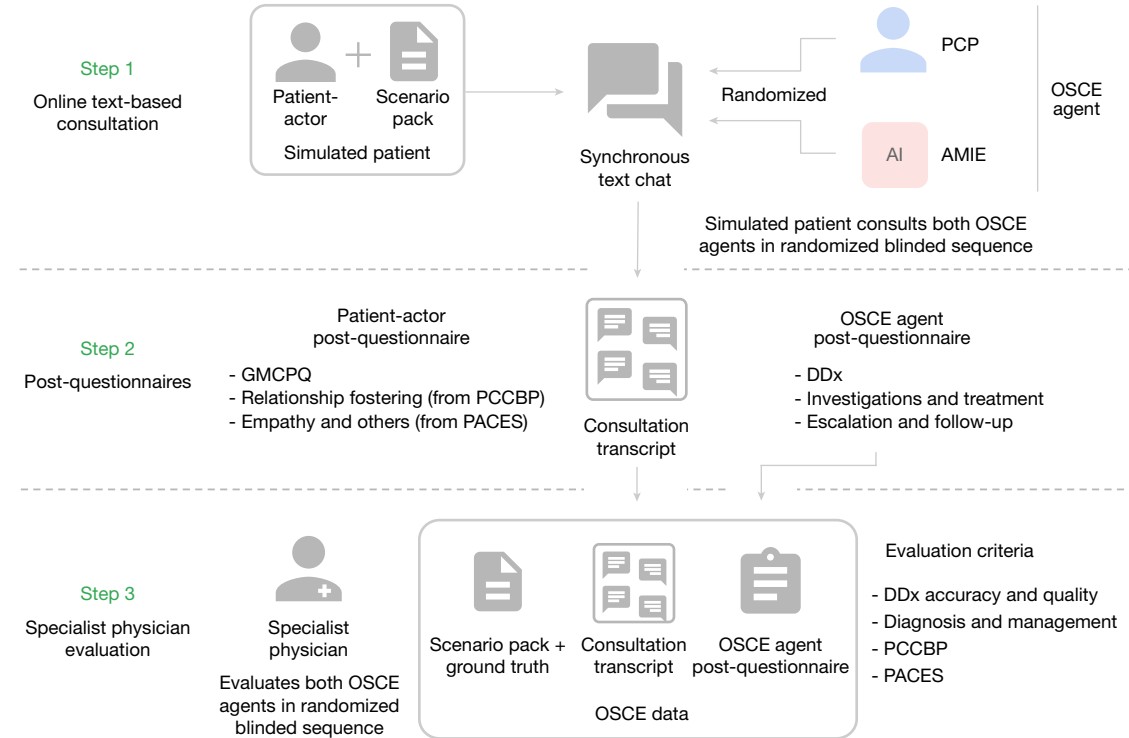

**Fig. 2 | Overview of randomized study design.** A PCP and AMIE perform (in a randomized order) a virtual remote OSCE with simulated patients by means of an online multi-turn synchronous text chat and produce answers to a post-questionnaire. Both the PCP and AMIE are then evaluated by both the patient-actors and specialist physicians.

in Canada, the United Kingdom and India, enabling the randomized and counterbalanced comparison of AMIE to primary care physicians (PCPs) when performing consultations with validated patient-actors. AMIE exhibited superior diagnostic accuracy compared to the PCPs, as assessed by various measures (for example, top-1 and top-3 accuracy of the differential diagnosis (DDx) list). Across 30 out of 32 evaluation axes from the specialist physician perspective and 25 out of 26 evaluation axes from the patient-actor perspective, AMIE was rated superior to PCPs while being non-inferior on the rest. Finally we performed a range of ablations to further understand and characterize the capabilities of AMIE, highlighting important limitations, and have proposed key next steps for the real-world clinical translation of AMIE.

Our research has important limitations, most notably that we utilized a text-chat interface, which, although enabling potentially large-scale interaction between patients and LLMs specialized for diagnostic dialogue, was unfamiliar to the PCPs for remote consultation. Thus, our study should not be regarded as representative of usual practice in (tele)medicine.

## Differential diagnosis accuracy
### AMIE has higher differential diagnosis accuracy than PCPs
AMIE's diagnostic accuracy was assessed as higher than that of the PCPs. Figure 3 shows the top-$k$ accuracy for AMIE and the PCPs, considering matches with the ground-truth diagnosis (Fig. 3a) and matches with any item on the accepted differential (Fig. 3b). AMIE showed significantly higher top-$k$ accuracy than that of the PCPs across all values of $k$ ($P < 0.05$). Note that, unlike AMIE, the PCPs did not always provide ten diagnoses in their DDxs (min = 3, mean = 5.36). Additionally, we performed a comparison of DDx accuracy between AMIE and the PCPs by varying the criteria for determining a match (that is, requiring an exact match versus just a highly relevant diagnosis). The results depicted in Supplementary Fig. 2 further substantiate AMIE's superior DDx performance across various matching criteria.

**Non-disease-state and disease-state accuracy.** Ten of the scenarios performed by AMIE and the PCPs were designed to primarily describe patients with no new concerning diagnosis (for example, a ground-truth diagnosis of resolved constipation, or the recurrence of a prior-known disease state of gastroesophageal-reflux-disease-induced chest pain). These were two scenarios each from the cardiovascular, gastroenterology, internal medicine, neurology and respiratory specialties. Here we plotted the top-$k$ DDx accuracy, as rated by the majority vote of three specialists for these non-disease-state cases. Although our results are not statistically significant, as they only consist of ten scenarios, AMIE appears to maintain the same trend of better performance on these mostly negative scenarios (Extended Data Fig. 2). AMIE has superior DDx accuracy on the set of 149 primarily positive disease state scenarios (in which only three scenarios had a ground-truth of a non-disease state).

**Accuracy by specialty.** Extended Data Fig. 3 illustrates the DDx accuracy achieved by AMIE and the PCPs across the six medical specialties covered by the scenarios in our study. We observed that AMIE's performance matched or surpassed PCP performance for all specialties except for obstetrics and gynaecology/urology, with the most pronounced improvements being in the respiratory and internal medicine specialties.

**Accuracy by location.** We observed that both AMIE and the PCPs had higher diagnostic accuracy in consultations performed in the Canada OSCE lab compared to those enacted in the India OSCE lab. However, the differences were not statistically significant and, in a subset of 40 scenarios enacted in both the Canada and India OSCE labs, the performances of both AMIE and the PCPs were equivalent (Extended Data Fig. 4).

## Efficiency in acquiring information
**Auto-evaluation accuracy.** We reproduced the DDx accuracy analysis with our model-based DDx auto-evaluator using the same procedure as in Fig. 3. The overall performance trends obtained through the

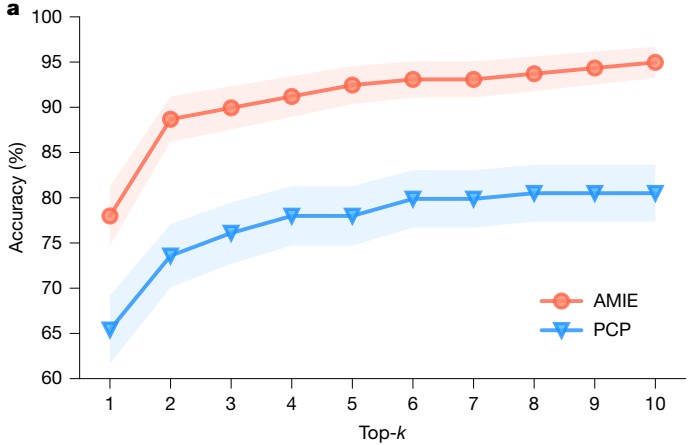

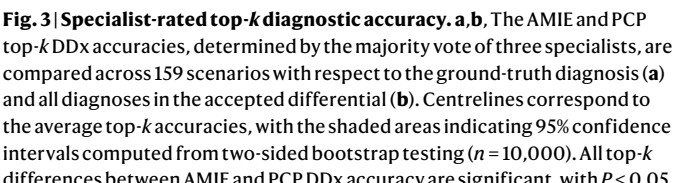

**Fig. 3 | Specialist-rated top-*k* diagnostic accuracy. a,b**, The AMIE and PCP top-*k* DDx accuracies, determined by the majority vote of three specialists, are compared across 159 scenarios with respect to the ground-truth diagnosis (**a**) and all diagnoses in the accepted differential (**b**). Centrelines correspond to the average top-*k* accuracies, with the shaded areas indicating 95% confidence intervals computed from two-sided bootstrap testing (*n* = 10,000). All top-*k* differences between AMIE and PCP DDx accuracy are significant, with *P* < 0.05

after FDR correction. The FDR-adjusted *P* values for ground-truth comparison are: 0.0017 (*k* = 1), 0.0002 (*k* = 2), 0.0002 (*k* = 3), 0.0002 (*k* = 4), 0.0002 (*k* = 5), 0.0003 (*k* = 6), 0.0003 (*k* = 7), 0.0003 (*k* = 8), 0.0002 (*k* = 9) and 0.0002 (*k* = 10) (**a**). The FDR-adjusted *P* values for accepted differential comparison are: 0.0001 (*k* = 1), 0.0001 (*k* = 2), 0.0002 (*k* = 3), 0.0002 (*k* = 4), 0.0001 (*k* = 5), 0.0001 (*k* = 6), 0.0001 (*k* = 7), 0.0001 (*k* = 8), 0.0001 (*k* = 9) and 0.0001 (*k* = 10) (**b**).

auto-evaluator align well with specialist assessments despite marginal differences in the computed accuracy values, as shown in Extended Data Fig. 5a,b. Additionally, we present a fully-simulated ablation testing different patient behaviours (Supplementary Fig. 3), which showed that AMIE was robust to many different patient personalities, although it had reduced DDx performance when interviewing patients with low English literacy.

**Isolating the source of performance gains.** To investigate whether AMIE's superior DDx performance observed in Fig. 3 stemmed from improved information acquisition or from better diagnostic reasoning capability, we compared AMIE's diagnoses based on its own consultations with AMIE's diagnoses generated from the corresponding PCP consultations, using the DDx auto-evaluator. The results depicted in Extended Data Fig. 5c,d revealed markedly similar DDx performance, indicating that the diagnostic performance remained consistent regardless of whether AMIE processed information from its own dialogue or from the PCP's conversation. Both methods significantly outperformed the DDxs produced by the PCPs. These results suggest that AMIE was approximately equivalent to the PCPs at information acquisition, but better than the PCPs at interpreting that information to produce an accurate or complete DDx.

**Efficiency of information acquisition.** Although AMIE displayed greater verbosity compared to the PCPs, in terms of total number of words generated in their responses during the consultation, the number of conversational turns and the number of words elicited from the patient-actors were similar across both OSCE agents, as illustrated in Extended Data Fig. 6a–c. This suggests that both AMIE and the PCPs acquired a similar amount of information from the patients during the encounter. To investigate how efficient AMIE or the PCPs were at gathering sufficient information to formulate a correct diagnosis, we truncated the conversations at various turn counts and used AMIE to generate DDxs based on these partial conversations. The results in Extended Data Fig. 6d,e illustrate that both AMIE and the PCPs were able to acquire the information necessary for formulating an accurate differential in the early stages (first ten turns) of the conversation. With comparable performance at all conversation lengths, neither AMIE nor the PCPs seemed to have a significant advantage in the speed, efficiency or diagnostic utility of information acquisition.

## Conversation quality
### AMIE surpasses PCPs in dialogue quality
Conversation quality was assessed using patient-actor ratings, specialist ratings and outputs from auto-evaluation. Supplementary Table 5 shows two example consultations with the same simulated patient from AMIE and a PCP.

**Patient-actor ratings.** Figure 4 presents the various conversation qualities the patient-actors assessed following their consultations with the OSCE agents. Overall, AMIE's consultations were rated significantly better (*P* < 0.05) by the patient-actors than those with the PCPs across 25 of 26 axes. No significant differences in ratings were detected for one of the patient-centred communication best practice (PCCBP) axes[19], 'Acknowledging mistakes' (*N* = 46). For this criterion, the number of exclusions was substantially higher because the question applied only when mistakes were made by the OSCE agent and were pointed out in the conversation.

**Specialist physician ratings.** Specialist physicians evaluated both the conversational quality as well as the responses to the post-questionnaire for scenarios within their domain expertise (Fig. 5). Again, AMIE's responses were rated significantly better by the specialists than those from the PCPs on 30 out of 32 evaluation axes, with the specialists preferring AMIE's consultations, diagnoses and management plans over those from the PCPs. For this set of evaluations, the differences in specialist ratings between AMIE and the PCPs were statistically significant (*P* < 0.05). See Supplementary Information section 7 for the inter-rater reliability between the three specialist raters per scenario. No significant differences in ratings were detected for two of the axes in the Diagnosis and management rubric—namely, 'Escalation recommendation appropriate' and 'Confabulation absent'—despite no exclusions (*N* = 159).

### Simulated dialogue conversation quality
We leveraged a model-based self-chain-of-thought auto-evaluation strategy (Supplementary Table 2) to rate conversations on four evaluation axes from the Practical Assessment of Clinical Examination Skills (PACES) rubric[20], and validated that these auto-evaluation ratings were accurate and well aligned with the specialist ratings (Supplementary

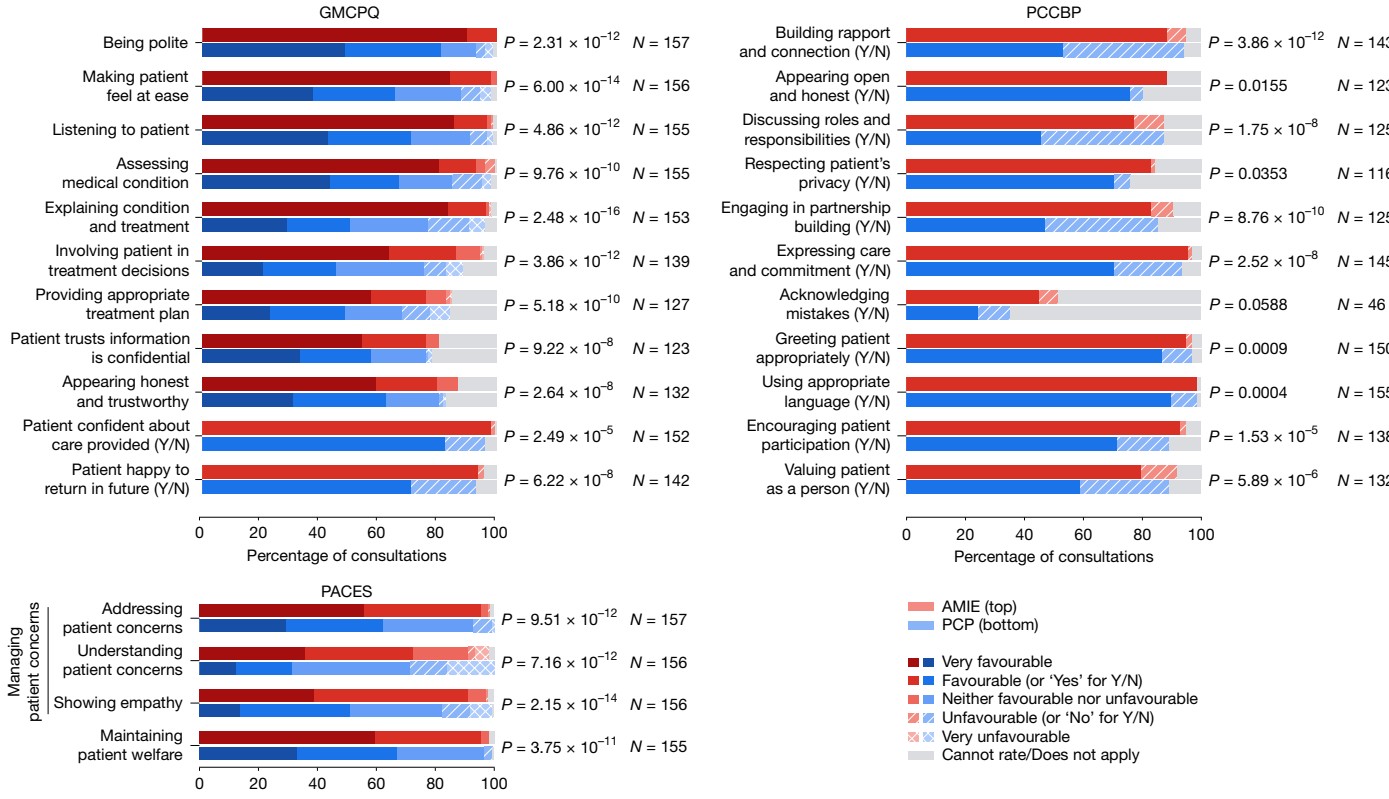

**Fig. 4 | Patient-actor ratings.** Conversation qualities, as assessed by the patient-actors upon conclusion of the consultation. For illustration purposes, all responses from the five-point rating scales were mapped to a generic five-point scale ranging from 'Very favourable' to 'Very unfavourable'. For Yes/No (Y/N) questions, a (positive) 'Yes' response was mapped to the same colour as 'Favourable' and a (negative) 'No' response to the same colour as 'Unfavourable'. The rating scales were adapted from the GMCPQ, PACES and a narrative review about PCCBP. Details on question-wording and response options are provided in Extended Data Tables 1 and 2. The evaluation involved 159 simulated patients. The *P* values were determined using two-sided Wilcoxon signed-rank tests with FDR correction. Cases where either AMIE or the PCP received 'Cannot rate/Does not apply' were excluded from the test.

Fig. 1b). Comparing the simulated dialogues generated before and after the self-play procedure, we found that the inner self-play loop improved simulated dialogue quality on these axes, as indicated in Supplementary Fig. 1c.

## Discussion

In this study, we introduced AMIE, an LLM-based AI system optimized for clinical dialogue with diagnostic reasoning capabilities. We compared AMIE consultations to those performed by PCPs using a randomized, double-blind crossover study with human simulated patients in the style of an OSCE. Notably, our study was not designed to be representative of clinical conventions either for traditional OSCE evaluations, for remote- or telemedical consultation practices or for the ways clinicians usually use text and chat messaging to communicate with patients. Our evaluation instead mirrored the most common way by which people interact with LLMs today, leveraging a potentially scalable and familiar mechanism for AI systems to engage in remote diagnostic dialogue. In this setting, we observed that AMIE, an AI system optimized specifically for the task, outperformed the PCPs on simulated diagnostic conversations when evaluated along multiple clinically meaningful axes of consultation quality.

### Diagnostic performance

The DDxs provided by AMIE were more accurate and complete than those provided by the board-certified PCPs when both were evaluated by specialist physicians. Previous research has shown that AI systems may match or exceed human diagnostic performance in specific, narrow tasks[21,22] in retrospective evaluation. However, these situations typically involved both the AI and physicians interpreting the same fixed input (for example, identifying the presence of a specific finding in a medical image). Our study was significantly more challenging because it required the AI system to actively acquire relevant information through conversation, rather than relying on clinical information collated by human efforts[23]. Therefore the system's downstream DDxs depended on not only its diagnostic inference capability, but also the quality of information gathered under uncertainty through natural conversation and building rapport.

Our results suggested that AMIE was as adept as the PCPs in eliciting pertinent information during the simulated consultations, and was more accurate than the PCPs in formulating a complete DDx if given the same amount of acquired information. This finding corroborates other work that LLMs may be able to produce more complete DDxs given the same clinical information as physicians in challenging cases[22]. Although not explored in this study, the assistive performance of AMIE therefore represents an interesting and important avenue for future research, particularly given the real-world importance of expert oversight for AI systems in safety-critical settings, such as medicine.

Our study utilized a wide variety of simulated patients, comprising actors trained in both Canada and India, and scenarios across a range of specialties. This allowed us to explore how performance varied along multiple axes—by specialty, and by the locations in which the scenario was derived and enacted. While we observed that both the PCPs and AMIE performed worse in gastroenterology and internal medicine scenarios than with other specialties (Extended Data Fig. 3), the study was not powered or designed to compare performance between different specialty topics and locations, and we cannot exclude that the scenarios in some specialties might have been harder than others.

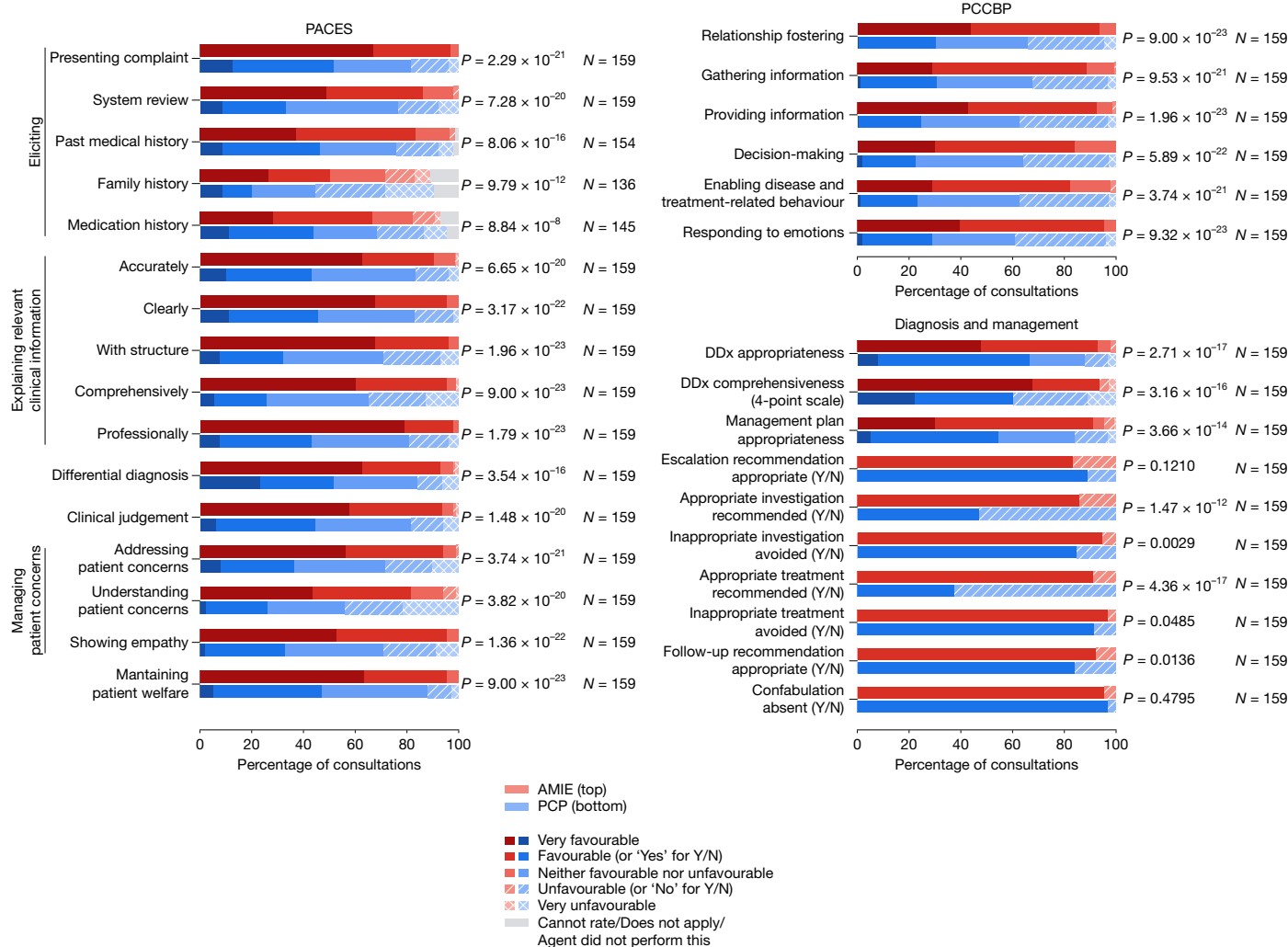

**Fig. 5 | Specialist physician ratings.** Conversation and reasoning qualities, as assessed by specialist physicians. For illustration purposes, all responses from the five-point rating scales were mapped to a generic five-point scale ranging from 'Very favourable' to 'Very unfavourable'. The only four-point scale (DDx comprehensiveness) was mapped to the same scale, ignoring the 'Neither favourable nor unfavourable' option. For Yes/No questions, a (positive) 'Yes' response was mapped to the same colour as 'Favourable' and a (negative) 'No' response to the same colour as 'Unfavourable'. The rating scales were adapted from PACES, a narrative review about PCCBP and other sources. Details on question-wording and response options are provided in Extended Data Tables 1–3. The evaluation involved 159 simulated patients, with the ratings from three distinct specialist physician raters for each case being aggregated using the median. The $P$ values were determined using two-sided Wilcoxon signed-rank tests with FDR correction. Cases where either AMIE or the PCP received 'Cannot rate/Does not apply' were excluded from the test.

## Conversational performance

The patient-actors and specialist raters both evaluated AMIE's performance to be higher than that of the PCPs on metrics related to empathy and communication skills. These axes comprised a majority of the dimensions that were evaluated. This general finding is consistent with a prior study, where LLM responses were found to be more empathetic than the responses from clinicians to health questions posted on Reddit[24]. However, the findings in that study cannot be generalized directly to our setting due to the differences in study design. Specifically, prior work has not involved a direct, randomized comparison of physicians and AI systems in a prospective simulation of multi-turn dialogue with the same patient. In both settings, the lack of voice-based and non-verbal visual communication may have been an unfair disadvantage to the clinicians.

The text-based chat interface used in this study introduced both advantages and disadvantages. People today most commonly engage with LLMs through synchronous text-chat interfaces[25], and patients often use patient portals to send messages to their providers. We therefore chose this mode of interaction as a representative interface for LLMs to perform multi-turn conversation, adapting the virtual OSCE framework accordingly. While this allowed a fair comparison of diagnostic dialogue between the LLMs and the clinicians when both were restricted to a synchronous text chat, it is important to acknowledge that our experiments did not emulate the expected quality of diagnostic dialogue in real clinical practice (including telemedicine). Physicians may be more used to history-taking and diagnostic dialogue by telephone or video consultation than synchronous text-chat communication[26]. Instead, text is more commonly used by clinicians to communicate with patients for episodic or asynchronous needs, such as prescription refills or communication about specific test results[27]. Physicians may thus be more familiar with text/SMS or email rather than the synchronous text-chat medium we employed in this study. In both text/SMS and email, the conventions and expectations for communicating naturally and with empathic style might be different[28]. It is possible that the PCPs in our study had not yet become accustomed to the setting, and may have performed differently if subjected to a specific training programme (similar in spirit to the training process for AMIE). Clinicians participating in the study undertook two

preparatory pilot sessions of consultations with our synchronous text interface before the evaluation began, but this was not a formal training programme, nor was it designed to optimize the clinicians' performance. Future research could explore this question more thoroughly, including monitoring for the impact of a learning curve or exploring whether performance varies according to the extent to which participating clinicians or simulated patients are familiar with telemedicine. Note that the conversations in our study were time-limited to follow typical OSCE conventions. While real-world patient–physician consultations often also take place under time constraints, the specific time limit imposed in our study may not be reflective of real-world scenarios.

Additionally, our findings regarding empathic communication could also be partially attributed to the fact that the AMIE responses were significantly longer than the clinician responses (Extended Data Fig. 6), and presented with greater structure. This could potentially suggest to an observer that more time was spent preparing the response, analogous to known findings that patient satisfaction increases with time spent with their physicians[29].

Collectively, our findings suggest many avenues for further research that might leverage human–AI complementarity[30], combining clinicians' skills in the analysis of verbal and non-verbal cues with the potential strengths of LLMs to suggest more enriched conversational responses, including empathic statements, structure, eloquence or more complete DDxs.

### Simulated dialogue

The use of simulated data allowed us to quickly scale the training to a broad set of conditions and patient contexts, while the injection of knowledge from search encouraged these dialogues to remain grounded and realistic. Although the simulated patients encompassed a wide range of conditions, they failed to capture the full range of potential patient backgrounds, personalities and motivations. Indeed, the simulated experiments shown in Supplementary Fig. 3 suggested that, while AMIE appears robust to certain variations in patient characteristics and behaviour, it has significant difficulty with some types of patients, such as those with low English literacy. Through the inner self-play procedure, we were able to iteratively improve the simulated dialogue we generated and used in fine-tuning. However, these improvements were limited by our ability to articulate what made good dialogue in the critic instructions, the critic's ability to produce effective feedback and AMIE's ability to adapt to such feedback. For example, in the simulated environment we imposed that AMIE reaches a proposed differential and testing/treatment plan for the patient, but such an endpoint may be unrealistic for some conditions, especially in the virtual chat-based setting. This limitation also applies in the real-world setting.

Additionally, the task of producing reward signals for the quality of medical diagnostic conversations is more challenging than evaluating outcomes in rule-based constrained environments where success is well-defined (for example, winning or losing a game of Go[31]). Our process for generating synthetic vignettes was designed with this consideration in mind. Because we knew the ground-truth condition for each vignette and the corresponding simulated dialogue(s) rollout, we were able to automatically assess the correctness of AMIE's DDx predictions as a proxy reward signal. This reward signal was used to filter out 'unsuccessful' simulated dialogues, such as those for which AMIE failed to produce an accurate DDx prediction during this self-play process. Beyond DDx accuracy, the self-play critic agent also assessed other qualities, including the level of empathy, professionalism and coherence conveyed by the doctor agent for each simulated dialogue. While these latter constructs are more subjective compared to diagnostic accuracy, they served as domain-specific heuristics imposed by clinical experts from our research team to help steer AMIE's development towards alignment with established clinical values. We also note that, in our preliminary analysis described in this work, our auto-evaluation framework for assessing the conversations along such rubrics was found to be in good alignment with human ratings and comparable to the inter-specialist agreement on these criteria.

Note that the majority of scenarios in our evaluation set assumed an underlying disease state, while only a small subset assumed the absence of disease. This is an important limitation of this work because it does not reflect the population-level epidemiological realities of primary care, where the majority of work in assessing patients involves ruling out disease, rather than ruling it in. We encourage future work to explore evaluation with various distributions of disease versus non-disease states.

Therefore, even within the distribution of diseases and specialties we addressed, our findings should be interpreted with humility and caution. There is a need for further research to examine varied presentations of the same diseases, alongside an exploration of alternative approaches to evaluating history-taking and clinical dialogue in situations of different patient needs, preferences, behaviours and circumstances.

### Fairness and bias

The evaluation protocol presented in this paper was limited in terms of its ability to capture potential issues related to fairness and bias, which remains an important open question that we will aim to address in subsequent system evaluations. Recent advances in the development of comprehensive frameworks for bias detection in LLMs[32] present a promising starting point for establishing such an approach. It should be noted that medical diagnostic dialogue is a particularly challenging use case, due to the complexity of the medical domain, the interactive information-gathering nature of the dialogue and the outcome-driven setting, with the potential of associated harms in cases of incorrect diagnosis or incorrect medical advice. Nevertheless, disentangling these issues is an important further research area if LLMs in the domain are to overcome, rather than propagate, inequities in healthcare. For example, previous studies have found that physicians approach communication with their patients differently, on average, depending on the patients' race, resulting in Black patients receiving communication that was less patient-centred and had a lower positive affect[33]. Other studies have found differences in physicians' communication styles and conversation length based on gender[34] and on patients' level of health literacy[35]. Effective intercultural communication skills are essential[36]. There is therefore a non-negligible risk that such historical conversational biases may be replicated or amplified in an AI dialogue system, but at the same time, there is also an opportunity to work towards designing conversational systems that can be more inclusive, and more personalized to the individual patient's needs.

To help inform the development of the necessary fairness, bias and equity frameworks, it was important to employ a participatory approach to solicit representative views across a wide range of patient demographics, as well as clinical and health equity domain experts. Such evaluation frameworks should be complemented by extensive model red-teaming and an adversarial approach to identifying any remaining gaps and failure modes. Recent advances in red-teaming LLMs could be useful in this scenario[37], where human raters or other AI systems (that is, the red team) simulate the role of an adversary to identify vulnerabilities and security gaps in these LLMs. These practices should not only inform the evaluation of the final model, but also its development and iterative refinement. Model development should follow the established data and model reporting practices and provide transparency into the training data and the associated decision processes[38–40]. The dialogue research dataset contributing to the AMIE training data in our study was de-identified, reducing the availability of socioeconomic factors, patient demographics and information about clinical settings and locations. To mitigate the risk that our synthetic vignettes would skew towards certain demographic groups, we leveraged web search to retrieve a range of demographics and associated symptoms relevant to each condition. We used these as input to the prompt template for vignette generation, instructing the model to produce multiple different vignettes given this range of inputs. While this mechanism was

designed with the intent of mitigating risks of bias amplification, a comprehensive evaluation of conversational diagnostic models, such as AMIE, for equity, fairness and bias is an important scope for future work.

Further work is also needed to ensure the robustness of medical LLMs in multilingual settings[41], and particularly their performance in minority languages[42]. The great variety of cultures[43], languages, localities, identities and localized medical needs makes the task of generating a priori static yet comprehensive fairness benchmarks practically infeasible. The measurement and mitigation of bias must move beyond the traditional narrow focus on specific axes that fails to scale globally[44]. With LLM-based evaluators, a potential solution is presented for preliminary assessments in languages where there are no systematic benchmarks, although prior studies have found these auto-evaluation frameworks to be biased, underscoring the need for calibrating them on native speaker evaluations, and using them with caution[45].

## Deployment

This study demonstrates the potential of LLMs for future use in healthcare in the context of diagnostic dialogue. Transitioning from an LLM research prototype that has been evaluated in this study to a safe and robust tool that can be used by healthcare providers, administrators and people will require significant additional research to ensure the safety, reliability, efficacy and privacy of the technology. Careful consideration will need to be given to the ethical deployment of this technology, including rigorous quality assessment across different clinical settings and research into reliable uncertainty estimation methods[46] that would allow for deferral to human clinical experts when needed. These and other guardrails are needed to mitigate the potential overreliance on LLM technologies, with other specific measures for attention to ethical and regulatory requirements particular to future use cases and the presence of qualified physicians in the loop to safeguard any model outputs. Additional research will also be needed to assess the extent to which biases and security vulnerabilities might arise, either from base models or the circumstances of use in deployment, as we have highlighted in our prior work[12]. Given the continuous evolution of clinical knowledge, it will also be important to develop ways for LLMs to utilize up-to-date clinical information[47].

## Conclusion

The utility of medical AI systems could be greatly improved if they are better able to interact conversationally, anchoring on large-scale medical knowledge, while communicating with appropriate levels of empathy and trust. This work demonstrates the great potential capabilities of LLM-based AI systems for settings involving clinical history-taking and diagnostic dialogue. The performance of AMIE in simulated consultations represents a milestone for the field, given it was assessed along an evaluation framework that considered multiple clinically relevant axes for conversational diagnostic medical AI. However, the results should be interpreted with appropriate caution. Translating from this limited scope of experimental simulated history-taking and diagnostic dialogue towards real-world tools for people and those who provide care for them requires a substantial amount of additional research and development to ensure the safety, reliability, fairness, efficacy and privacy of the technology. If successful, we believe AI systems, such as AMIE, can be at the core of next-generation-learning health systems that help scale world-class healthcare to everyone.

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

## Methods

### Real-world datasets for AMIE

AMIE was developed using a diverse suite of real-world datasets, including multiple-choice medical question-answering, expert-curated long-form medical reasoning, electronic health record (EHR) note summaries and large-scale transcribed medical conversation interactions. As described in detail below, in addition to dialogue generation tasks, the training task mixture for AMIE consisted of medical question-answering, reasoning and summarization tasks.

**Medical reasoning.** We used the MedQA (multiple-choice) dataset, consisting of US Medical Licensing Examination multiple-choice-style open-domain questions with four or five possible answers[48]. The training set consisted of 11,450 questions and the test set had 1,273 questions. We also curated 191 MedQA questions from the training set where clinical experts had crafted step-by-step reasoning leading to the correct answer[13].

**Long-form medical question-answering.** The dataset used here consisted of expert-crafted long-form responses to 64 questions from HealthSearchQA, LiveQA and Medication QA in MultiMedQA[12].

**Medical summarization.** A dataset consisting of 65 clinician-written summaries of medical notes from MIMIC-III, a large, publicly available database containing the medical records of intensive care unit patients[49], was used as additional training data for AMIE. MIMIC-III contains approximately two million notes spanning 13 types, including cardiology, respiratory, radiology, physician, general, discharge, case management, consult, nursing, pharmacy, nutrition, rehabilitation and social work. Five notes from each category were selected, with a minimum total length of 400 tokens and at least one nursing note per patient. Clinicians were instructed to write abstractive summaries of individual medical notes, capturing key information while also permitting the inclusion of new informative and clarifying phrases and sentences not present in the original note.

**Real-world dialogue.** Here we used a de-identified dataset licensed from a dialogue research organization, comprising 98,919 audio transcripts of medical conversations during in-person clinical visits from over 1,000 clinicians over a ten-year period in the United States[50]. It covered 51 medical specialties (primary care, rheumatology, haematology, oncology, internal medicine and psychiatry, among others) and 168 medical conditions and visit reasons (type 2 diabetes, rheumatoid arthritis, asthma and depression being among the common conditions). Audio transcripts contained utterances from different speaker roles, such as doctors, patients and nurses. On average, a conversation had 149.8 turns ($P_{0.25} = 75.0$, $P_{0.75} = 196.0$). For each conversation, the metadata contained information about patient demographics, reason for the visit (follow-up for pre-existing condition, acute needs, annual exam and more), and diagnosis type (new, existing or other unrelated). Refer to ref. 50 for more details.

For this study, we selected dialogues involving only doctors and patients, but not other roles, such as nurses. During preprocessing, we removed paraverbal annotations, such as '[LAUGHING]' and '[INAUDIBLE]', from the transcripts. We then divided the dataset into training (90%) and validation (10%) sets using stratified sampling based on condition categories and reasons for visits, resulting in 89,027 conversations for training and 9,892 for validation.

### Simulated learning through self-play

While passively collecting and transcribing real-world dialogues from in-person clinical visits is feasible, two substantial challenges limit its effectiveness in training LLMs for medical conversations: (1) existing real-world data often fail to capture the vast range of medical conditions and scenarios, hindering its scalability and comprehensiveness; and (2) the data derived from real-world dialogue transcripts tend to be noisy, containing ambiguous language (including slang, jargon and sarcasm), interruptions, ungrammatical utterances and implicit references. This, in turn, may have limited AMIE's knowledge, capabilities and applicability.

To address these limitations, we designed a self-play-based simulated learning environment for diagnostic medical dialogues in a virtual care setting, enabling us to scale AMIE's knowledge and capabilities across a multitude of medical conditions and contexts. We used this environment to iteratively fine-tune AMIE with an evolving set of simulated dialogues in addition to the static corpus of medical question-answering, reasoning, summarization and real-world dialogue data described above.

This process consisted of two self-play loops:
- An inner self-play loop where AMIE leveraged in-context critic feedback to refine its behaviour on simulated conversations with an AI patient agent.
- An outer self-play loop where the set of refined simulated dialogues were incorporated into subsequent fine-tuning iterations. The resulting new version of AMIE could then participate in the inner loop again, creating a continuous learning cycle.

At each iteration of fine-tuning, we produced 11,686 dialogues, stemming from 5,230 different medical conditions. The conditions were selected from three datasets:
- The Health QA dataset[12], which contained 613 common medical conditions.
- The MalaCards Human Disease Database (https://github.com/ Shivanshu-Gupta/web-scrapers/blob/master/medical_ner/malacards-diseases.json), which contained 18,455 less-common disease conditions.
- The MedicineNet Diseases & Conditions Index (https://github. com/Shivanshu-Gupta/web-scrapers/blob/master/medical_ner/ medicinenet-diseases.json), which contained 4,617 less-common conditions.

At each self-play iteration, four conversations were generated from each of the 613 common conditions, while two conversations were generated from each of the 4,617 less-common conditions randomly chosen from MedicineNet and MalaCards. The average simulated dialogue conversation length was 21.28 turns ($P_{0.25} = 19.0$, $P_{0.75} = 25.0$).

**Simulated dialogues through self-play.** To produce high-quality simulated dialogues at scale, we developed a new multi-agent framework that comprised three key components:
- A vignette generator: AMIE leverages web searches to craft unique patient vignettes given a specific medical condition.
- A simulated dialogue generator: three LLM agents play the roles of patient agent, doctor agent and moderator, engaging in a turn-by-turn dialogue simulating realistic diagnostic interactions.
- A self-play critic: a fourth LLM agent acts as a critic to give feedback to the doctor agent for self-improvement. Notably, AMIE acted as all agents in this framework.

The prompts for each of these steps are listed in Supplementary Table 3. The vignette generator aimed to create varied and realistic patient scenarios at scale, which could be subsequently used as context for generating simulated doctor–patient dialogues, thereby allowing AMIE to undergo a training process emulating exposure to a greater number of conditions and patient backgrounds. The patient vignette (scenario) included essential background information, such as patient demographics, symptoms, past medical history, past surgical history, past social history and patient questions, as well as an associated diagnosis and management plan.

For a given condition, patient vignettes were constructed using the following process. First, we retrieved 60 passages (20 each) on

the range of demographics, symptoms and management plans associated with the condition from using an internet search engine. To ensure these passages were relevant to the given condition, we used the general-purpose LLM, PaLM 2 (ref. 10), to filter these retrieved passages, removing any passages deemed unrelated to the given condition. We then prompted AMIE to generate plausible patient vignettes aligned with the demographics, symptoms and management plans retrieved from the filtered passages, by providing a one-shot exemplar to enforce a particular vignette format.

Given a patient vignette detailing a specific medical condition, the simulated dialogue generator was designed to simulate a realistic dialogue between a patient and a doctor in an online chat setting where in-person physical examination may not be feasible.

Three specific LLM agents (patient agent, doctor agent and moderator), each played by AMIE, were tasked with communicating among each other to generate the simulated dialogues. Each agent had distinct instructions. The patient agent embodied the individual experiencing the medical condition outlined in the vignette. Their role involved truthfully responding to the doctor agent's inquiries, as well as raising any additional questions or concerns they may have had. The doctor agent played the role of an empathetic clinician seeking to comprehend the patient's medical history within the online chat environment[51]. Their objective was to formulate questions that could effectively reveal the patient's symptoms and background, leading to an accurate diagnosis and an effective treatment plan. The moderator continually assessed the ongoing dialogue between the patient agent and doctor agent, determining when the conversation had reached a natural conclusion.

The turn-by-turn dialogue simulation started with the doctor agent initiating the conversation: "Doctor: So, how can I help you today?". Following this, the patient agent responded, and their answer was incorporated into the ongoing dialogue history. Subsequently, the doctor agent formulated a response based on the updated dialogue history. This response was then appended to the conversation history. The conversation progressed until the moderator detected the dialogue had reached a natural conclusion, when the doctor agent had provided a DDx, treatment plan, and adequately addressed any remaining patient agent questions, or if either agent initiated a farewell.

To ensure high-quality dialogues, we implemented a tailored self-play[3,52] framework specifically for the self-improvement of diagnostic conversations. This framework introduced a fourth LLM agent to act as a 'critic', which was also played by AMIE, and that was aware of the ground-truth diagnosis to provide in-context feedback to the doctor agent and enhance its performance in subsequent conversations.

Following the critic's feedback, the doctor agent incorporated the suggestions to improve its responses in subsequent rounds of dialogue with the same patient agent from scratch. Notably, the doctor agent retained access to its previous dialogue history in each new round. This self-improvement process was repeated twice to generate the dialogues used for each iteration of fine-tuning. See Supplementary Table 4 as an example of this self-critique process.

We noted that the simulated dialogues from self-play had significantly fewer conversational turns than those from the real-world data described in the previous section. This difference was expected, given that our self-play mechanism was designed—through instructions to the doctor and moderator agents—to simulate text-based conversations. By contrast, real-world dialogue data was transcribed from in-person encounters. There are fundamental differences in communication styles between text-based and face-to-face conversations. For example, in-person encounters may afford a higher communication bandwidth, including a higher total word count and more 'back and forth' (that is, a greater number of conversational turns) between the physician and the patient. AMIE, by contrast, was designed for focused information gathering by means of a text-chat interface.

## Instruction fine-tuning

AMIE, built upon the base LLM PaLM 2 (ref. 10), was instruction fine-tuned to enhance its capabilities for medical dialogue and reasoning. We refer the reader to the PaLM 2 technical report for more details on the base LLM architecture. Fine-tuning examples were crafted from the evolving simulated dialogue dataset generated by our four-agent procedure, as well as the static datasets. For each task, we designed task-specific instructions to instruct AMIE on what task it would be performing. For dialogue, this was assuming either the patient or doctor role in the conversation, while for the question-answering and summarization datasets, AMIE was instead instructed to answer medical questions or summarize EHR notes. The first round of fine-tuning from the base LLM only used the static datasets, while subsequent rounds of fine-tuning leveraged the simulated dialogues generated through the self-play inner loop.

For dialogue generation tasks, AMIE was instructed to assume either the doctor or patient role and, given the dialogue up to a certain turn, to predict the next conversational turn. When playing the patient agent, AMIE's instruction was to reply to the doctor agent's questions about their symptoms, drawing upon information provided in patient scenarios. These scenarios included patient vignettes for simulated dialogues or metadata, such as demographics, visit reason and diagnosis type, for the real-world dialogue dataset. For each fine-tuning example in the patient role, the corresponding patient scenario was added to AMIE's context. In the doctor agent role, AMIE was instructed to act as an empathetic clinician, interviewing patients about their medical history and symptoms to ultimately arrive at an accurate diagnosis. From each dialogue, we sampled, on average, three turns for each doctor and patient role as the target turns to predict based on the conversation leading up to that target turn. Target turns were randomly sampled from all turns in the dialogue that had a minimum length of 30 characters.

Similarly, for the EHR note summarization task, AMIE was provided with a clinical note and prompted to generate a summary of the note. Medical reasoning/QA and long-form response generation tasks followed the same set-up as in ref. 13. Notably, all tasks except dialogue generation and long-form response generation incorporated few-shot (1–5) exemplars in addition to task-specific instructions for additional context.

## Chain-of-reasoning for online inference

To address the core challenge in diagnostic dialogue—effectively, acquiring information under uncertainty to enhance diagnostic accuracy and confidence, while maintaining positive rapport with the patient—AMIE employed a chain-of-reasoning strategy before generating a response in each dialogue turn. Here 'chain-of-reasoning' refers to a series of sequential model calls, each dependent on the outputs of prior steps. Specifically, we used a three-step reasoning process, described as follows:

- Analysing patient information. Given the current conversation history, AMIE was instructed to: (1) summarize the positive and negative symptoms of the patient as well as any relevant medical/family/social history and demographic information; (2) produce a current DDx; (3) note missing information needed for a more accurate diagnosis; and (4) assess confidence in the current differential and highlight its urgency.
- Formulating response and action. Building upon the conversation history and the output of step 1, AMIE: (1) generated a response to the patient's last message and formulated further questions to acquire missing information and refine the DDx; and (2) if necessary, recommended immediate action, such as an emergency room visit. If confident in the diagnosis, based on the available information, AMIE presented the differential.
- Refining the response. AMIE revised its previous output to meet specific criteria based on the conversation history and outputs from earlier steps. The criteria were primarily related to factuality and

formatting of the response (for example, avoid factual inaccuracies on patient facts and unnecessary repetition, show empathy, and display in a clear format).

This chain-of-reasoning strategy enabled AMIE to progressively refine its response conditioned on the current conversation to arrive at an informed and grounded reply.

## Evaluation

Prior works developing models for clinical dialogue have focused on metrics, such as the accuracy of note-to-dialogue or dialogue-to-note generations[53,54], or natural language generation metrics, such as BLEU or ROUGE scores that fail to capture the clinical quality of a consultation[55,56].

In contrast to these prior works, we sought to anchor our human evaluation in criteria more commonly used for evaluating the quality of physicians' expertise in history-taking, including their communication skills in consultation. Additionally, we aimed to evaluate conversation quality from the perspective of both the lay participant (the participating patient-actor) and a non-participating professional observer (a physician who was not directly involved in the consultation). We surveyed the literature and interviewed clinicians working as OSCE examiners in Canada and India to identify a minimum set of peer-reviewed published criteria that they considered comprehensively reflected the criteria that are commonly used in evaluating both patient-centred and professional-centred aspects of clinical diagnostic dialogue—that is, identifying the consensus for PCCBP in medical interviews[19], the criteria examined for history-taking skills by the Royal College of Physicians in the United Kingdom as part of their PACES (https://www.mrcpuk. org/mrcpuk-examinations/paces/marksheets)[20] and the criteria proposed by the UK GMCPQ (https://edwebcontent.ed.ac.uk/sites/default/files/imports/fileManager/patient_questionnaire%20pdf_48210488. pdf) for doctors seeking patient feedback as part of professional revalidation (https://www.gmc-uk.org/registration-and-licensing/managing-your-registration/revalidation/revalidation-resources).

The resulting evaluation framework enabled assessment from two perspectives—the clinician, and lay participants in the dialogues (that is, the patient-actors). The framework included the consideration of consultation quality, structure and completeness, and the roles, responsibilities and skills of the interviewer (Extended Data Tables 1–3).

**Remote OSCE study design.** To compare AMIE's performance to that of real clinicians, we conducted a randomized crossover study of blinded consultations in the style of a remote OSCE. Our OSCE study involved 20 board-certified PCPs and 20 validated patient-actors, ten each from India and Canada, respectively, to partake in online text-based consultations (Extended Data Fig. 1). The PCPs had between 3 and 25 years of post-residency experience (median 7 years). The patient-actors comprised of a mix of medical students, residents and nurse practitioners with experience in OSCE participation. We sourced 159 scenario packs from India (75), Canada (70) and the United Kingdom (14).

The scenario packs and simulated patients in our study were prepared by two OSCE laboratories (one each in Canada and India), each affiliated with a medical school and with extensive experience in preparing scenario packs and simulated patients for OSCE examinations. The UK scenario packs were sourced from the samples provided on the Membership of the Royal Colleges of Physicians UK website. Each scenario pack was associated with a ground-truth diagnosis and a set of acceptable diagnoses. The scenario packs covered conditions from the cardiovascular (31), respiratory (32), gastroenterology (33), neurology (32), urology, obstetric and gynaecology (15) domains and internal medicine (16). The scenarios are listed in Supplementary Information section 8. The paediatric and psychiatry domains were excluded from this study, as were intensive care and inpatient case management scenarios.

Indian patient-actors played the roles in all India scenario packs and 7 of the 14 UK scenario packs. Canadian patient-actors participated in

scenario packs for both Canada and the other half of the UK-based scenario packs. This assignment process resulted in 159 distinct simulated patients (that is, scenarios). Below, we use the term 'OSCE agent' to refer to the conversational counterpart interviewing the patient-actor—that is, either the PCP or AMIE. Supplementary Table 1 summarizes the OSCE assignment information across the three geographical locations. Each of the 159 simulated patients completed the three-step study flow depicted in Fig. 2.

**Online text-based consultation.** The PCPs and patient-actors were primed with sample scenarios and instructions, and participated in pilot consultations before the study began to familiarize them with the interface and experiment requirements.

For the experiment, each simulated patient completed two online text-based consultations by means of a synchronous text-chat interface (Extended Data Fig. 1), one with a PCP (control) and one with AMIE (intervention). The ordering of the PCP and AMIE was randomized and the patient-actors were not informed as to which they were talking to in each consultation (counterbalanced design to control for any potential order effects). The PCPs were located in the same country as the patient-actors, and were randomly drawn based on availability at the time slot specified for the consultation. The patient-actors role-played the scenario and were instructed to conclude the conversation after no more than 20 minutes. Both OSCE agents were asked (the PCPs through study-specific instructions and AMIE as part of the prompt template) to not reveal their identity, or whether they were human, under any circumstances.

**Post-questionnaires.** Upon conclusion of the consultation, the patient-actor and OSCE agent each filled in a post-questionnaire in light of the resulting consultation transcript (Extended Data Fig. 1). The post-questionnaire for patient-actors consisted of the complete GMCPQ, the PACES components for 'Managing patient concerns' and 'Maintaining patient welfare' (Extended Data Table 1) and a checklist representation of the PCCBP category for 'Fostering the relationship' (Extended Data Table 2). The responses the patient-actors provided to the post-questionnaire are referred to as 'patient-actor ratings'. The post-questionnaire for the OSCE agent asked for a ranked DDx list with a minimum of three and no more than ten conditions, as well as recommendations for escalation to in-person or video-based consultation, investigations, treatments, a management plan and the need for a follow-up.

**Specialist physician evaluation.** Finally, a pool of 33 specialist physicians from India (18), North America (12) and the United Kingdom (3) evaluated the PCPs and AMIE with respect to the quality of their consultation and their responses to the post-questionnaire. During evaluation, the specialist physicians also had access to the full scenario pack, along with its associated ground-truth differential and additional accepted differentials. All of the data the specialist physicians had access to during evaluation are collectively referred to as 'OSCE data'. Specialist physicians were sourced to match the specialties and geographical regions corresponding to the scenario packs included in our study, and had between 1 and 32 years of post-residency experience (median 5 years). Each set of OSCE data was evaluated by three specialist physicians randomly assigned to match the specialty and geographical region of the underlying scenario (for example, Canadian pulmonologists evaluated OSCE data from the Canada-sourced respiratory medicine scenario). Each specialist evaluated the OSCE data from both the PCP and AMIE for each given scenario. Evaluations for the PCP and AMIE were conducted by the same set of specialists in a randomized and blinded sequence.

Evaluation criteria included the accuracy, appropriateness and comprehensiveness of the provided DDx list, the appropriateness of recommendations regarding escalation, investigation, treatment, management plan and follow-up (Extended Data Table 3) and all PACES

(Extended Data Table 1) and PCCBP (Extended Data Table 2) rating items. We also asked specialist physicians to highlight confabulations in the consultations and questionnaire responses—that is, text passages that were non-factual or that referred to information not provided in the conversation. Each OSCE scenario pack additionally supplied the specialists with scenario-specific clinical information to assist with rating the clinical quality of the consultation, such as the ideal investigation or management plans, or important aspects of the clinical history that would ideally have been elucidated for the highest quality of consultation possible. This follows the common practice for instructions for OSCE examinations, in which specific clinical scenario-specific information is provided to ensure consistency among examiners, and follows the paradigm demonstrated by Membership of the Royal Colleges of Physicians sample packs. For example, this scenario (https://www.thefederation.uk/sites/default/files/Station%202%20Scenario%20Pack%20%2816%29.pdf) informs an examiner that, for a scenario in which the patient-actor has haemoptysis, the appropriate investigations would include a chest X-ray, a high-resolution computed tomography scan of the chest, a bronchoscopy and spirometry, whereas bronchiectasis treatment options a candidate should be aware of should include chest physiotherapy, mucolytics, bronchodilators and antibiotics.

**Statistical analysis and reproducibility.** We evaluated the top-$k$ accuracy of the DDx lists generated by AMIE and the PCPs across all 159 simulated patients. Top-$k$ accuracy was defined as the percentage of cases where the correct ground-truth diagnosis appeared within the top-$k$ positions of the DDx list. For example, top-3 accuracy is the percentage of cases for which the correct ground-truth diagnosis appeared in the top three diagnosis predictions from AMIE or the PCP. Specifically, a candidate diagnosis was considered a match if the specialist rater marked it as either an exact match with the ground-truth diagnosis, or very close to or closely related to the ground-truth diagnosis (or accepted differential). Each conversation and DDx was evaluated by three specialists, and their majority vote or median rating was used to determine the accuracy and quality ratings, respectively.

The statistical significance of the DDx accuracy was determined using two-sided bootstrap tests[57] with 10,000 samples and false discovery rate (FDR) correction[58] across all $k$. The statistical significance of the patient-actor and specialist ratings was determined using two-sided Wilcoxon signed-rank tests[59], also with FDR correction. Cases where either agent received 'Cannot rate/Does not apply' were excluded from the test. All significance results are based on $P$ values after FDR correction.

Additionally, we reiterate that the OSCE scenarios themselves were sourced from three different countries, the patient-actors came from two separate institutions in Canada and India, and the specialist evaluations were triplicate rated in this study.

## Related work

**Clinical history-taking and the diagnostic dialogue.** History-taking and the clinical interview are widely taught in both medical schools and postgraduate curricula[60–65]. Consensus on physician–patient communication has evolved to embrace patient-centred communication practices, with recommendations that communication in clinical encounters should address six core functions—fostering the relationship, gathering information, providing information, making decisions, responding to emotions and enabling disease- and treatment-related behaviour[19,66,67]. The specific skills and behaviours for meeting these goals have also been described, taught and assessed[19,68] using validated tools[68]. Medical conventions consistently cite that certain categories of information should be gathered during a clinical interview, comprising topics such as the presenting complaint, past medical history and medication history, social and family history, and systems review[69,70]. Clinicians' ability to meet these goals is commonly assessed using the framework of an OSCE[4,5,71]. Such assessments vary in their reproducibility or implementation, and have even been adapted for remote practice as virtual OSCEs with telemedical scenarios, an issue of particular relevance during the COVID-19 pandemic[72].

**Conversational AI and goal-oriented dialogue.** Conversational AI systems for goal-oriented dialogue and task completion have a rich history[73–75]. The emergence of transformers[76] and large language models[15] have led to renewed interest in this direction. The development of strategies for alignment[77], self-improvement[78–81] and scalable oversight mechanisms[82] has enabled the large-scale deployment of such conversational systems in the real world[16,83]. However, the rigorous evaluation and exploration of conversational and task-completion capabilities of such AI systems remains limited for clinical applications, where studies have largely focused on single-turn interaction use cases, such as question-answering or summarization.

**AI for medical consultations and diagnostic dialogue.** The majority of explorations of AI as tools for conducting medical consultations have focused on 'symptom-checker' applications rather than a full natural dialogue, or on topics such as the transcription of medical audio or the generation of plausible dialogue, given clinical notes or summaries[84–87]. Language models have been trained using clinical dialogue datasets, but these have not been comprehensively evaluated[88,89]. Studies have been grounded in messages between doctors and patients in commercial chat platforms (which may have altered doctor–patient engagement compared to 1:1 medical consultations)[55,90,91]. Many have focused largely on predicting next turns in the recorded exchanges rather than clinically meaningful metrics. Also, to date, there have been no reported studies that have examined the quality of AI models for diagnostic dialogue using the same criteria used to examine and train human physicians in dialogue and communication skills, nor studies evaluating AI systems in common frameworks, such as the OSCE.

**Evaluation of diagnostic dialogue.** Prior frameworks for the human evaluation of AI systems' performance in diagnostic dialogue have been limited in detail. They have not been anchored in established criteria for assessing communication skills and the quality of history-taking. For example, ref. 56 reported a five-point scale describing overall 'human evaluation', ref . 90 reported 'relevance, informativeness and human likeness', and ref . 91 reported 'fluency, expertise and relevance', whereas other studies have reported 'fluency and adequacy'[92] and 'fluency and specialty'[93]. These criteria are far less comprehensive and specific than those taught and practiced by medical professionals. A multi-agent framework for assessing the conversational capabilities of LLMs was introduced in ref. 88, the study, however, was performed in the restricted setting of dermatology, used AI models to emulate both the doctor and patient sides of simulated interactions, and it performed limited expert evaluation of the history-taking as being complete or not.

## Reporting summary

Further information on research design is available in the Nature Portfolio Reporting Summary linked to this article.

## Data availability

Many of the real-world datasets used in the development of AMIE are open-source, including MedQA (https://github.com/jind11/MedQA), MultiMedQA (https://www.nature.com/articles/s41586-023-06291-2#data-availability) and MIMIC-III (https://physionet.org/content/mimiciii/1.4/). The scenario packs from the United Kingdom used in the OSCE study are also available for download from https://www.thefederation.uk/sites/default/files/documents/Station%202%20Scenario%20Pack%20%2816%29.pdf. Additional scenario packs used in the study will be made available upon request.

## Code availability

AMIE is an LLM-based research AI system for diagnostic dialogue. Reviewers were provided access to the system through a testing program to interact with the system and evaluate the performance. We are not open-sourcing model code and weights due to the safety implications of the unmonitored use of such a system in medical settings. In the interest of responsible innovation, we will be working with research partners, regulators and providers to validate and explore safe onward uses of AMIE. For reproducibility, we have documented technical deep-learning methods while keeping the paper accessible to a clinical and general scientific audience. Our work builds upon PaLM 2, for which technical details have been described extensively in the technical report[10]. All analyses were conducted using Python v.2.7.18 (https://www.python.org/).

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

**Acknowledgements** This project represents an extensive collaboration between several teams at Google Research and Google DeepMind. We thank Y. Liu, D. McDuff, J. Sunshine, A. Connell, P. McGovern and Z. Ghahramani for their comprehensive reviews and detailed feedback on early versions of the manuscript. We also thank S. Lachgar, L. Winer, J. Guilyard and M. Shiels for contributions to the narratives and visuals. We are grateful to J. A. Seguin, S. Goldman, Y. Vasilevski, X. Song, A. Goel, C.-l. Ko, A. Das, H. Yu, C. Liu, Y. Liu, S. Man, B. Hatfield, S. Li, A. Joshi, G. Turner, A. Um'rani, D. Pandya and P. Singh for their valuable insights, technical support and feedback during our research. We also thank GoodLabs Studio Inc., Intel Medical Inc. and C. Smith for their partnership in conducting the OSCE study in North America, and the JSS Academy of Higher Education and Research and V. Patil for their partnership in conducting the OSCE study in India. Finally, we are grateful to D. Webster, E. Dominowska, D. Fleet, P. Mansfield, S. Prakash, R. Wong, S. Thomas, M. Howell, K. DeSalvo, J. Dean, J. Manyika, Z. Ghahramani and D. Hassabis for their support during the course of this project.

**Author contributions** A.P., M.S., T.T., S.S.M., K. Singhal, S.A., A.K., R.T., J.F. and V.N. contributed to the conception and design of the work; A.P., M.S., T.T., S.S.M., K. Saab, A.K., A. Wang, K.K. and V.N. contributed to the data acquisition and curation; A.P., M.S., T.T., K. Saab, A.K., Y.C., R.T., J.F., N.T., E.V., B.L., M.A. and V.N. contributed to the technical implementation; A.K., V.N., M.S., T.T., A.P. and N.T. contributed to the evaluation framework used in the study; Y.C., L.H., A. Webson and J.G. provided technical and infrastructure guidance; A.K. provided clinical inputs to the study; C.S., J.G., J.B., K.C., G.S.C. and Y.M. contributed to the ideation and execution of the work. All authors contributed to the drafting and revising of the manuscript.

**Competing interests** This study was funded by Alphabet Inc. and/or a subsidiary thereof ('Alphabet'). All authors are employees of Alphabet and may own stock as part of the standard compensation package.

**Additional information**
**Correspondence and requests for materials** should be addressed to Tao Tu, Mike Schaekermann, Alan Karthikesalingam or Vivek Natarajan.

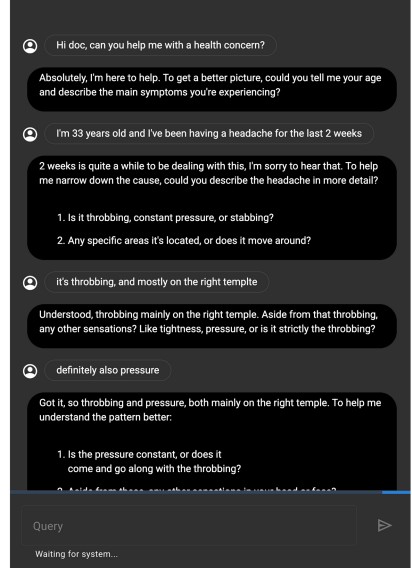

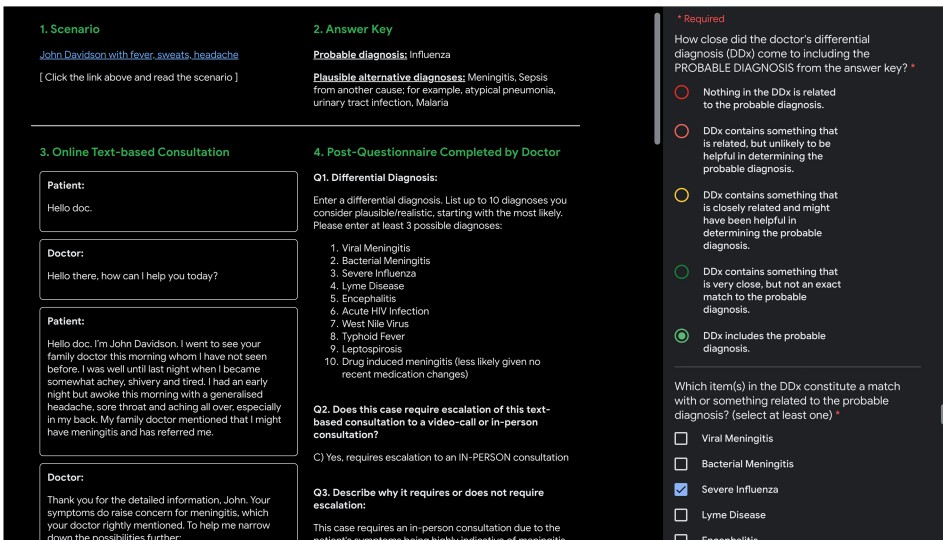

Chat Interface                    Specialist Physician Evaluation Interface

**Extended Data Fig. 1 | User interfaces for the online consultation and evaluation processes.** Online consultations between patient actors and either AMIE or the primary care physicians (PCPs) were conducted by means of a synchronous text-based chat interface. The evaluation process was facilitated through a rating interface in which specialist physicians were provided the scenario information including differential diagnosis answer key, as well as a consultation transcript along with post-questionnaire responses from AMIE or the PCPs. Rating prompts were provided alongside these pieces of information.

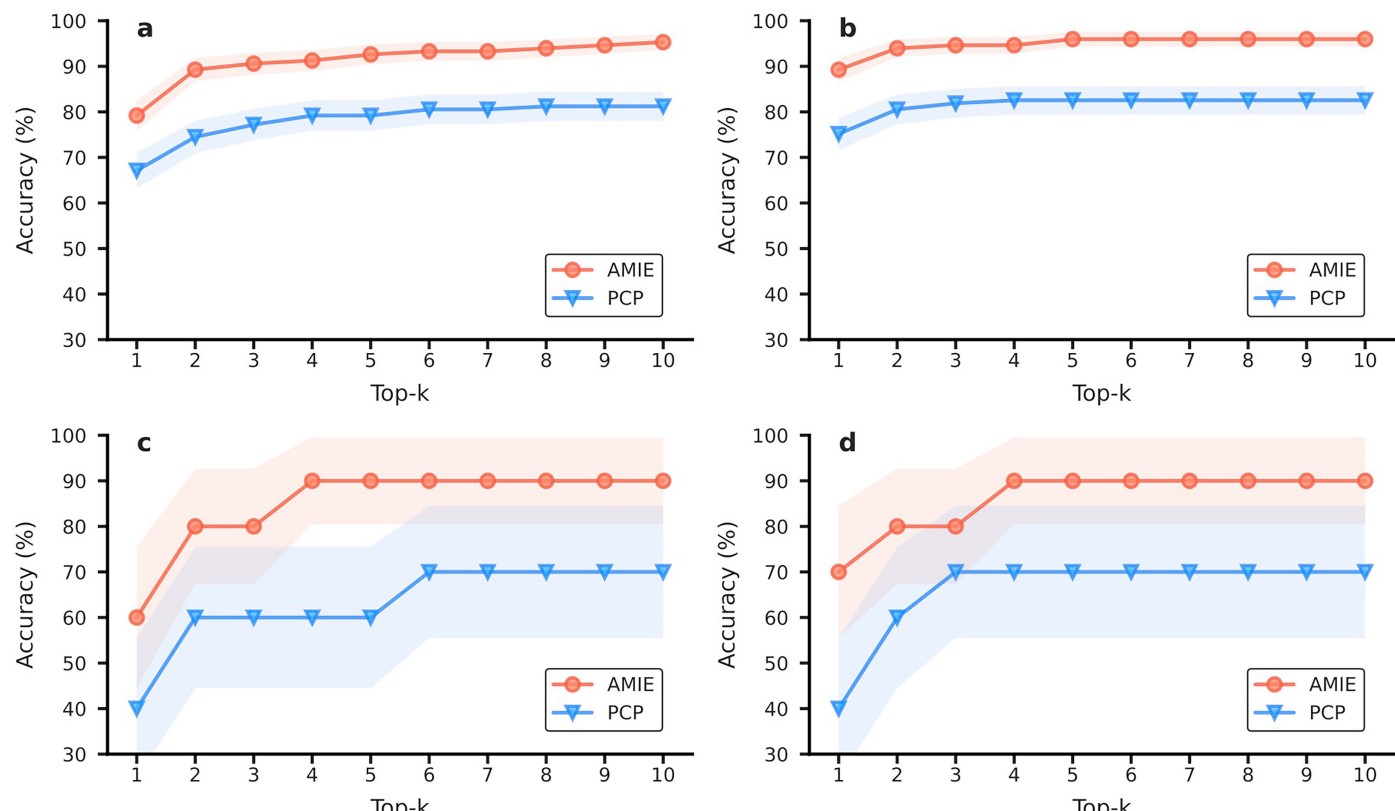

**Extended Data Fig. 2 | DDx top-*k* accuracy for non-disease-states and positive disease-states. a,b:** Specialist rated DDx top-*k* accuracy for the 149 "positive" scenarios with respect to (**a**) the ground-truth diagnosis and (**b**) the accepted differentials. **c,d:** Specialist rated DDx top-*k* accuracy for the 10 "negative" scenarios with respect to (**c**) the ground-truth diagnosis and (**d**) the accepted differentials. Using two-sided bootstrap tests (*n* = 10,000) with FDR correction, differences in the "positive" scenarios were significant (*P* < 0.05) for all k, but differences in "negative" scenarios were not significant due to the small sample size. Centrelines correspond to the average top-*k* accuracy, with 95% confidence intervals shaded. The FDR-adjusted *P* values for positive disease states, ground-truth comparison: 0.0041 (*k* = 1), 0.0002 (*k* = 2),

0.0001 (*k* = 3), 0.0002 (*k* = 4), 0.0001 (*k* = 5), 0.0002 (*k* = 6), 0.0002 (*k* = 7), 0.0003 (*k* = 8), 0.0001 (*k* = 9) and 0.0001 (*k* = 10) (**a**). The FDR-adjusted *P* values for positive disease states, accepted differential comparison: 0.0002 (*k* = 1), 0.0001 (*k* = 2), 0.0002 (*k* = 3), 0.0003 (*k* = 4), 0.0001 (*k* = 5), 0.0001 (*k* = 6), 0.0001 (*k* = 7), 0.0001 (*k* = 8), 0.0001 (*k* = 9) and 0.0001 (*k* = 10) (**b**). The FDR-adjusted *P* values for non-disease states, ground-truth comparison: 0.1907 (*k* = 1), 0.1035 (*k* = 2), 0.1035 (*k* = 3), 0.1035 (*k* = 4), 0.1035 (*k* = 5), 0.1035 (*k* = 6), 0.1035 (*k* = 7), 0.1035 (*k* = 8), 0.1035 (*k* = 9) and 0.1035 (*k* = 10) (**c**). The FDR-adjusted *P* values for non-disease states, accepted differential comparison: 0.1035 (*k* = 1), 0.1035 (*k* = 2), 0.1829 (*k* = 3), 0.1035 (*k* = 4), 0.1035 (*k* = 5), 0.1035 (*k* = 6), 0.1035 (*k* = 7), 0.1035 (*k* = 8), 0.1035 (*k* = 9) and 0.1035 (*k* = 10) (**d**).

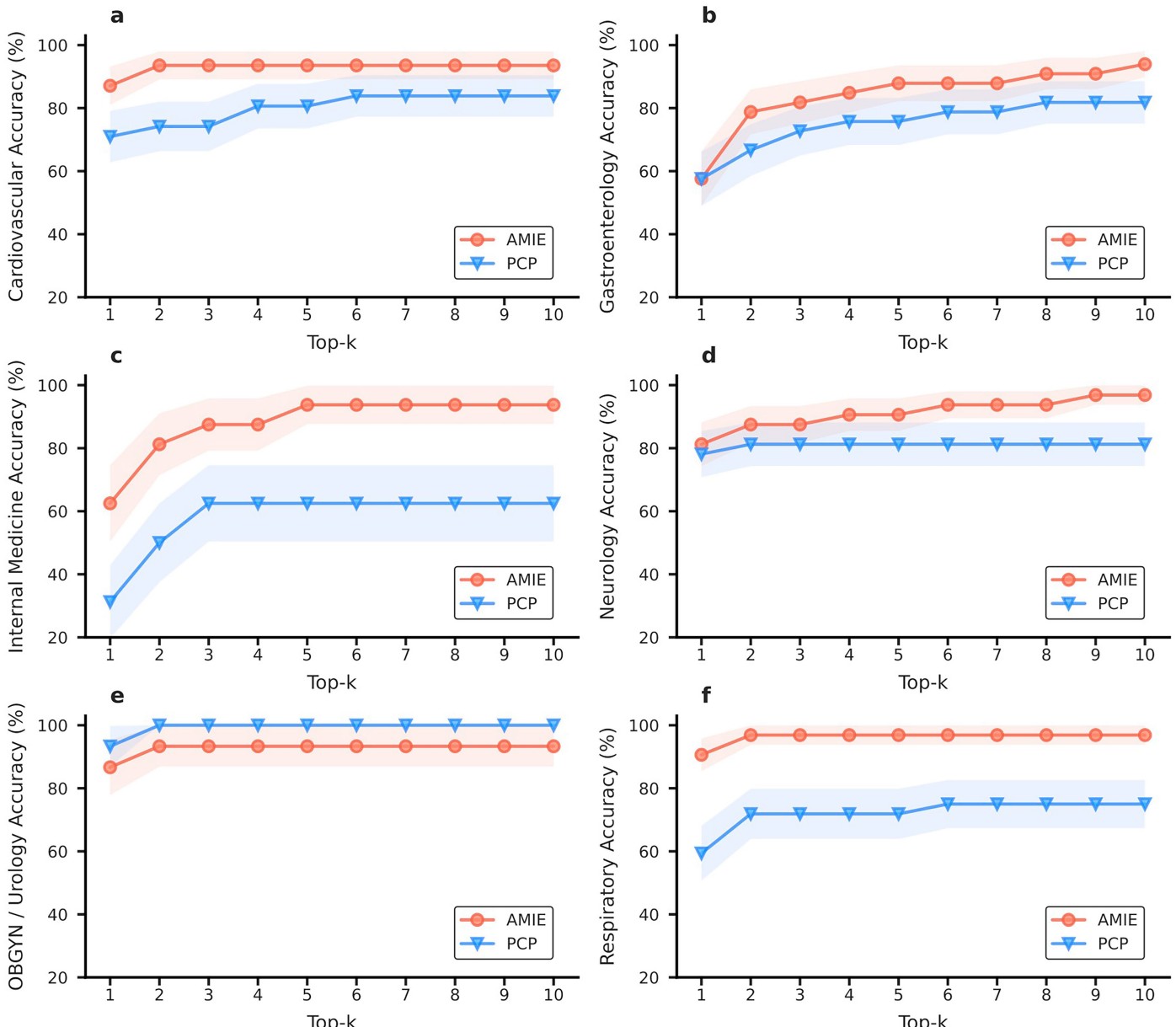

**Extended Data Fig. 3 | Specialist rated DDx accuracy by scenario specialty.**
Top-*k* DDx accuracy for scenarios with respect to the ground-truth in (**a**)
Cardiology (*N* = 31, not significant), (**b**) Gastroenterology (*N* = 33, not significant),
(**c**) Internal Medicine (*N* = 16, significant for all *k*), (**d**) Neurology (*N* = 32,
significant for k > 5), (**e**) Obstetrics and Gynaecology (OBGYN)/Urology (*N* = 15,
not significant), (**f**) Respiratory (*N* = 32, significant for all *k*). Two-sided bootstrap
tests (*n* = 10,000) with FDR correction were used to assess significance (*P* < 0.05)
on these cases. Centrelines correspond to the average top-*k* accuracy, with 95%
confidence intervals shaded. The FDR-adjusted *P* values for Cardiology: 0.0911
(*k* = 1), 0.0637 (*k* = 2), 0.0637 (*k* = 3), 0.0911 (*k* = 4), 0.0911 (*k* = 5), 0.0929 (*k* = 6),
0.0929 (*k* = 7), 0.0929 (*k* = 8), 0.0929 (*k* = 9) and 0.0929 (*k* = 10) (**a**). The
FDR-adjusted *P* values for Gastroenterology: 0.4533 (*k* = 1), 0.1735 (*k* = 2),
0.1735 (*k* = 3), 0.1735 (*k* = 4), 0.1735 (*k* = 5), 0.1735 (*k* = 6), 0.1735 (*k* = 7), 0.1735 (*k* = 8),
0.1735 (*k* = 9) and 0.1735 (*k* = 10) (**b**). The FDR-adjusted *P* values for Internal
Medicine: 0.0016 (*k* = 1), 0.0102 (*k* = 2), 0.0216 (*k* = 3), 0.0216 (*k* = 4), 0.0013 (*k* = 5),
0.0013 (*k* = 6), 0.0013 (*k* = 7), 0.0013 (*k* = 8), 0.0013 (*k* = 9) and 0.0013 (*k* = 10) (**c**).
The FDR-adjusted *P* values for Neurology: 0.2822 (*k* = 1), 0.1655 (*k* = 2), 0.1655
(*k* = 3), 0.069 (*k* = 4), 0.069 (*k* = 5), 0.0492 (*k* = 6), 0.0492 (*k* = 7), 0.0492 (*k* = 8),
0.0492 (*k* = 9) and 0.0492 (*k* = 10) (**d**). The FDR-adjusted *P* values for OBGYN/
Urology: 0.285 (*k* = 1), 0.1432 (*k* = 2), 0.1432 (*k* = 3), 0.1432 (*k* = 4), 0.1432 (*k* = 5),
0.1432 (*k* = 6), 0.1432 (*k* = 7), 0.1432 (*k* = 8), 0.1432 (*k* = 9) and 0.1432 (*k* = 10) (**e**).
The FDR-adjusted *P* values for Respiratory: 0.0004 (*k* = 1), 0.0004 (*k* = 2), 0.0004
(*k* = 3), 0.0004 (*k* = 4), 0.0004 (*k* = 5), 0.0006 (*k* = 6), 0.0006 (*k* = 7), 0.0006 (*k* = 8),
0.0006 (*k* = 9) and 0.0006 (*k* = 10) (**f**).

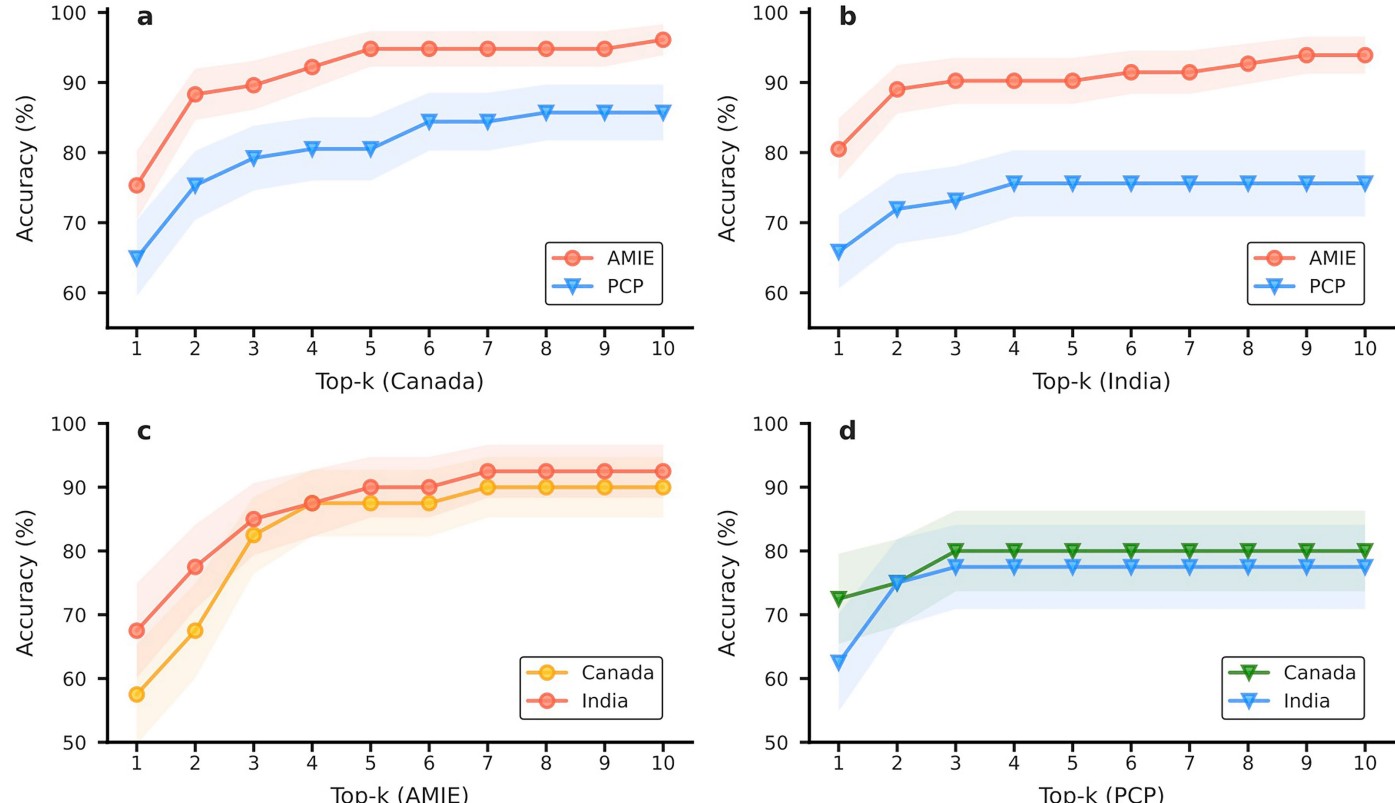

**Extended Data Fig. 4 | DDx accuracy by location. a, b:** Specialist DDx rating of AMIE and the PCPs with respect to the ground-truth for the 77 cases conducted in Canada (**a**) and 82 cases in India (**b**). The differences between AMIE and the PCPs performance are significant for all values of $k$. **c, d:** Auto-evaluation rated DDx for 40 scenarios which were duplicated in both Canada and India for AMIE (**c**) and the PCPs (**d**). The differences between Canada and India performance are not significant on these shared scenarios, for both AMIE and the PCPs. Significance was determined using two-sided bootstrap tests ($n$=10,000) with FDR correction. Centrelines correspond to the average top-$k$ accuracy, with 95% confidence intervals shaded. The FDR-adjusted $P$ values for Canada comparison: 0.0438 ($k$=1), 0.0289 ($k$=2), 0.0438 ($k$=3), 0.0305 ($k$=4), 0.0267 ($k$=5), 0.0267 ($k$=6), 0.0267 ($k$=7), 0.0305 ($k$=8), 0.0305 ($k$=9) and 0.0276 ($k$=10) (**a**). The FDR-adjusted $P$ values for India comparison: 0.0037 ($k$=1), 0.0005 ($k$=2), 0.0005 ($k$=3), 0.0013 ($k$=4), 0.0013 ($k$=5), 0.0009 ($k$=6), 0.0009 ($k$=7), 0.0005 ($k$=8), 0.0005 ($k$=9) and 0.0005 ($k$=10) (**b**). The FDR-adjusted $P$ values for shared AMIE scenarios: 0.3465 ($k$=1), 0.3465 ($k$=2), 0.4109 ($k$=3), 0.4109 ($k$=4), 0.3465 ($k$=5), 0.3465 ($k$=6), 0.3465 ($k$=7), 0.3465 ($k$=8), 0.3465 ($k$=9) and 0.3465 ($k$=10) (**c**). The FDR-adjusted $P$ values for shared PCP scenarios: 0.3905 ($k$=1), 0.4356 ($k$=2), 0.3905 ($k$=3), 0.3905 ($k$=4), 0.3905 ($k$=5), 0.3905 ($k$=6), 0.3905 ($k$=7), 0.3905 ($k$=8), 0.3905 ($k$=9) and 0.3905 ($k$=10) (**d**).

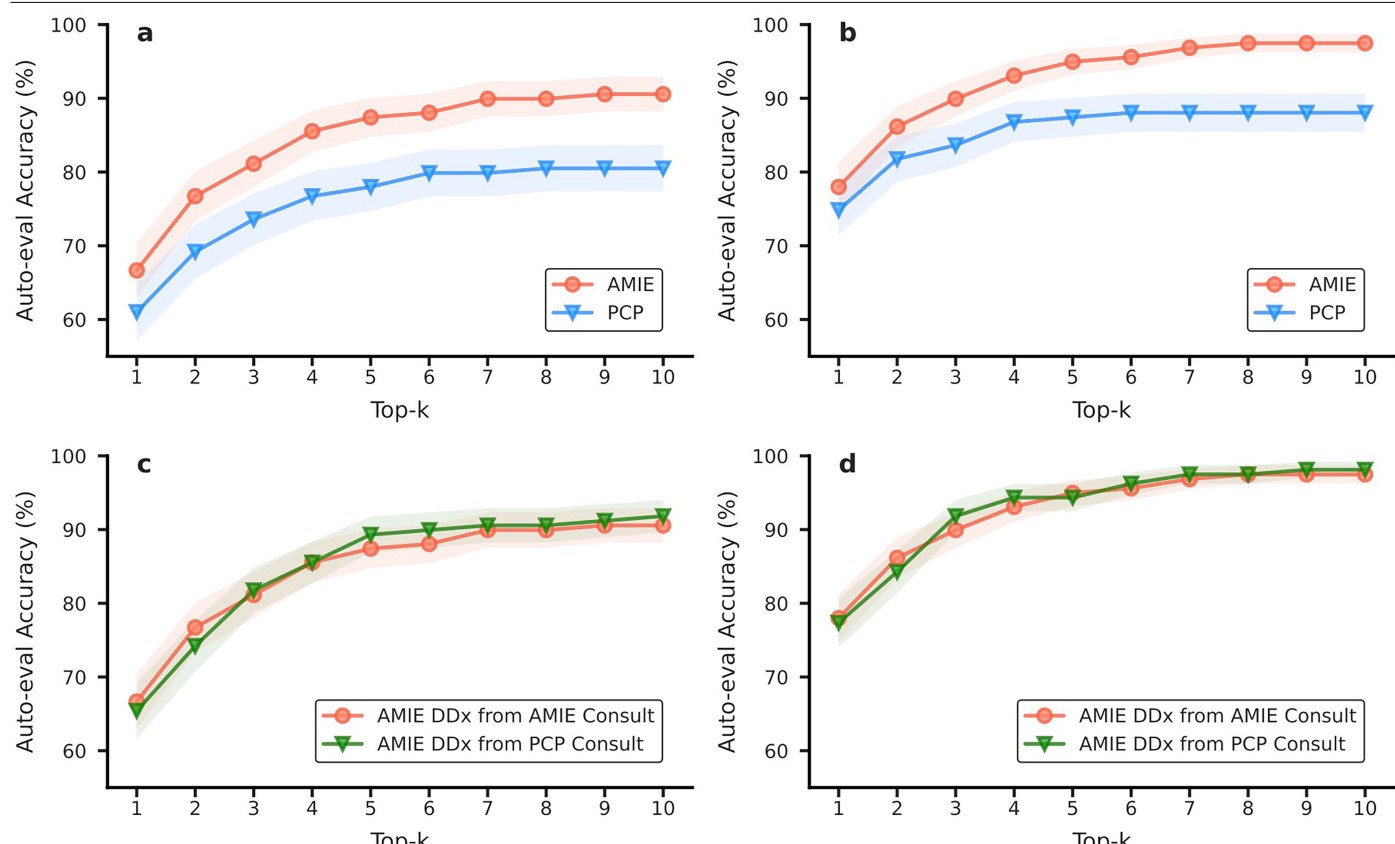

**Extended Data Fig. 5 | Auto-evaluation of DDx performance. a, b:** Top-*k* DDx auto-evaluation of AMIE's and the PCP's differential diagnoses from their own consultations with respect to the ground-truth (**a**, significant for *k* > 3) and the list of accepted differentials (**b**, significant for *k* > 4). **c, d:** Top-*k* DDx auto-evaluation of AMIE's differential diagnoses when provided its own vs. the PCP's consultation transcript with respect to the ground-truth (**c**, not significant) and the list of accepted differentials (**d**, not significant). Two-sided bootstrap tests (*n* = 10,000) with FDR correction were used to assess significance (*P* < 0.05) on these 159 cases. Centrelines correspond to the average top-*k* accuracy, with 95% confidence intervals shaded. The FDR-adjusted *P* values for AMIE vs. the PCP ground-truth comparison: 0.1399 (*k* = 1), 0.0737 (*k* = 2), 0.0596 (*k* = 3),

0.0315 (*k* = 4), 0.0221 (*k* = 5), 0.0315 (*k* = 6), 0.0182 (*k* = 7), 0.0221 (*k* = 8), 0.0182 (*k* = 9) and 0.0182 (*k* = 10) (**a**). The FDR-adjusted *P* values for AMIE vs. the PCP accepted differential comparison: 0.2297 (*k* = 1), 0.1713 (*k* = 2), 0.0779 (*k* = 3), 0.0546 (*k* = 4), 0.018 (*k* = 5), 0.0174 (*k* = 6), 0.006 (*k* = 7), 0.0033 (*k* = 8), 0.0033 (*k* = 9) and 0.0033 (*k* = 10) (**b**). The FDR-adjusted *P* values for AMIE vs. the PCP consultation ground-truth comparison: 0.4929 (*k* = 1), 0.4929 (*k* = 2), 0.4929 (*k* = 3), 0.4929 (*k* = 4), 0.4929 (*k* = 5), 0.4929 (*k* = 6), 0.4929 (*k* = 7), 0.4929 (*k* = 8), 0.4929 (*k* = 9) and 0.4929 (*k* = 10) (**c**). The FDR-adjusted *P* values for AMIE vs. the PCP consultation accepted differential comparison: 0.4461 (*k* = 1), 0.4461 (*k* = 2), 0.4461 (*k* = 3), 0.4461 (*k* = 4), 0.4461 (*k* = 5), 0.4461 (*k* = 6), 0.4461 (*k* = 7), 0.4461 (*k* = 8), 0.4461 (*k* = 9) and 0.4461 (*k* = 10) (**d**).

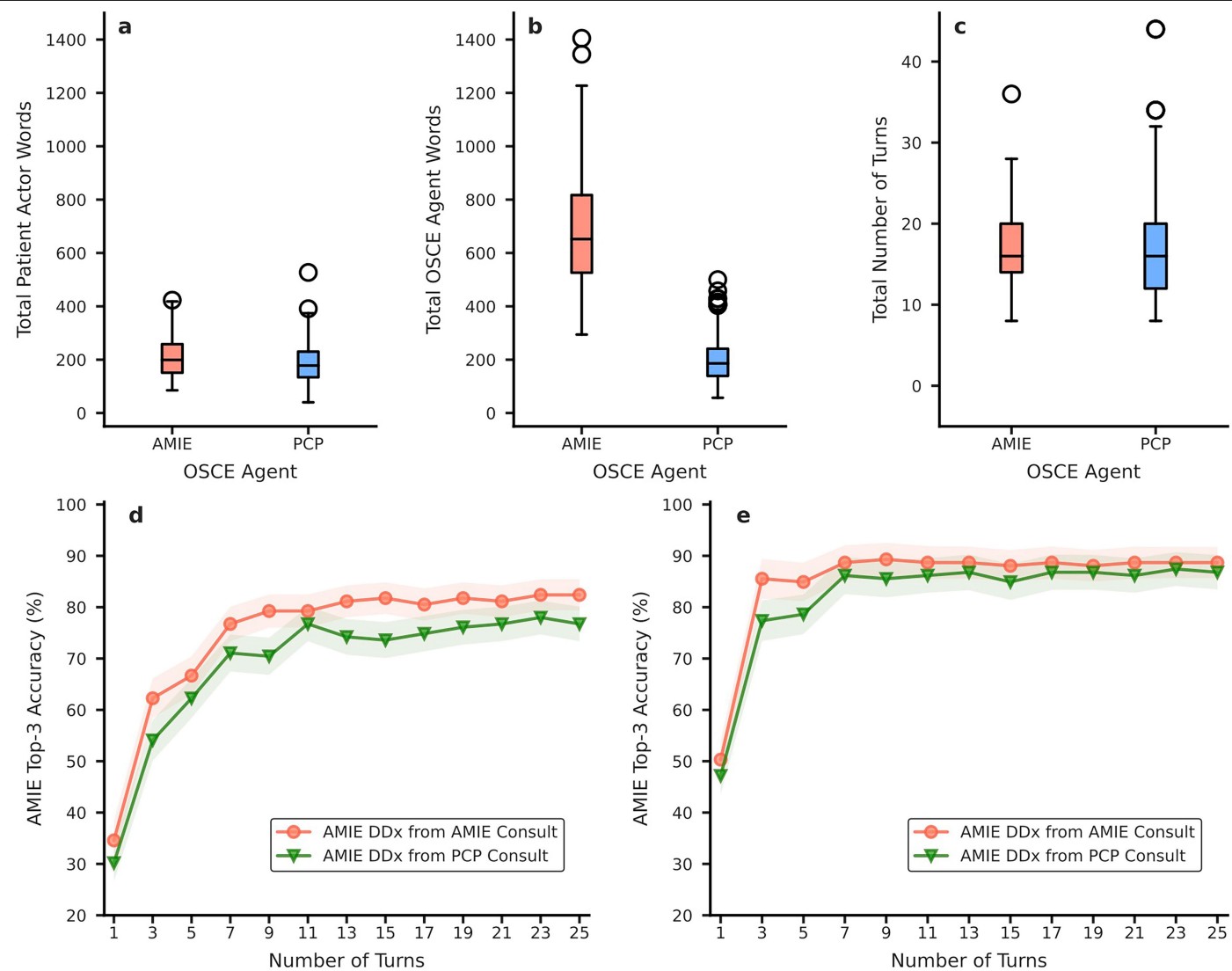

**Extended Data Fig. 6 | Consultation verbosity and efficiency of information acquisition. a**, Total patient actor words elicited by AMIE and the PCPs. **b**, Total words sent to patient actor from AMIE and the PCPs. **c**, Total number of turns in AMIE vs. the PCP consultations. For (**a-c**), Centrelines correspond to the median, with the box indicating 25th and 75th percentiles. The minimum and maximum are presented as the bottom and top whiskers, respectively, excluding the outliers which are defined as data points further than 1.5 times the inter-quartile range from the box. **d, e:** The top-3 auto-evaluation rated DDx accuracy of AMIE using the first *T* turns of each consultation, with respect to the ground-truth diagnosis (**d**) and the accepted differentials (**e**). Differences on these 159 cases are not significant (*P* > 0.05) when compared through two-sided bootstrap tests (*n* =10,000) with FDR correction. Centrelines correspond to the average top-3 accuracy, with 95% confidence intervals shaded.

**Extended Data Table 1 | Practical Assessment of Clinical Examination Skills (PACES) rubric details**

## Practical Assessment of Clinical Examination Skills (PACES)

| Question | Scale | Assessed by |
|---|---|---|
| **Clinical Communication Skills** | | |
| To what extent did the doctor elicit the PRESENTING COMPLAINT? | 5-point scale | Specialist |
| To what extent did the doctor elicit the SYSTEMS REVIEW? | 5-point scale | Specialist |
| To what extent did the doctor elicit the PAST MEDICAL HISTORY? | 5-point scale | Specialist |
| To what extent did the doctor elicit the FAMILY HISTORY? | 5-point scale | Specialist |
| To what extent did the doctor elicit the MEDICATION HISTORY? | 5-point scale | Specialist |
| To what extent did the doctor explain relevant clinical information ACCURATELY? | 5-point scale | Specialist |
| To what extent did the doctor explain relevant clinical information CLEARLY? | 5-point scale | Specialist |
| To what extent did the doctor explain relevant clinical information WITH STRUCTURE? | 5-point scale | Specialist |
| To what extent did the doctor explain relevant clinical information COMPREHENSIVELY? | 5-point scale | Specialist |
| To what extent did the doctor explain relevant clinical information PROFESSIONALLY? | 5-point scale | Specialist |
| **Differential Diagnosis** | | |
| To what extent did the doctor construct a sensible DIFFERENTIAL DIAGNOSIS? | 5-point scale | Specialist |
| **Clinical Judgement** | | |
| To what extent did the doctor select a comprehensive, sensible and appropriate MANAGEMENT PLAN? | 5-point scale | Specialist |
| **Managing Patient Concerns** | | |
| To what extent did the doctor seek, detect, acknowledge and attempt to address the patient's concerns? | 5-point scale | Specialist & Patient Actor |
| To what extent did the doctor confirm the patient's knowledge and understanding? | 5-point scale | Specialist & Patient Actor |
| How empathic was the doctor? | 5-point scale | Specialist & Patient Actor |
| **Maintaining Patient Welfare** | | |
| To what extent did the doctor maintain the patient's welfare? | 5-point scale | Specialist & Patient Actor |

**Extended Data Table 2 | Patient-Centred Communication Best Practice (PCCBP) rubric details**

## Patient-Centered Communication Best Practice (PCCBP)

| Question | Scale | Assessed by |
|---|---|---|
| **Fostering the Relationship** | | |
| How would you rate the doctor's behavior of FOSTERING A RELATIONSHIP with the patient? | 5-point scale | Specialist |
| | Binary scale per criterion | Patient Actor |
| **Gathering Information** | | |
| How would you rate the doctor's behavior of GATHERING INFORMATION from the patient? | 5-point scale | Specialist |
| **Providing Information** | | |
| How would you rate the doctor's behavior of PROVIDING INFORMATION to the patient? | 5-point scale | Specialist |
| **Decision Making** | | |
| How would you rate the doctor's behavior of MAKING DECISIONS with the patient? | 5-point scale | Specialist |
| **Enabling Disease and Treatment-Related Behavior** | | |
| How would you rate the doctor's behavior of ENABLING DISEASE AND TREATMENT-RELATED BEHAVIOR in the patient? | 5-point scale | Specialist |
| **Responding to Emotions** | | |
| How would you rate the doctor's behavior of RESPONDING TO EMOTIONS expressed by the patient? | 5-point scale | Specialist |

**Extended Data Table 3 | Diagnosis and Management rubric details**

### Diagnosis & Management

| Question | Scale | Options | Assessed by |
|---|---|---|---|
| **Diagnosis** | | | |
| How APPROPRIATE was the doctor's differential diagnosis (DDx) compared to the answer key? | 5-point scale | Very Inappropriate<br>Inappropriate<br>Neither Appropriate Nor Inappropriate<br>Appropriate<br>Very Appropriate | Specialist |
| How COMPREHENSIVE was the doctor's differential diagnosis (DDx) compared to the answer key? | 4-point scale | The DDx has major candidates missing.<br>The DDx contains some of the candidates but a number are missing.<br>The DDx contains most of the candidates but some are missing.<br>The DDx contains all candidates that are reasonable. | Specialist |
| How close did the doctor's differential diagnosis (DDx) come to including the PROBABLE DIAGNOSIS from the answer key? | 5-point scale | Nothing in the DDx is related to the probable diagnosis.<br>DDx contains something that is related, but unlikely to be helpful in determining the probable diagnosis.<br>DDx contains something that is closely related and might have been helpful in determining the probable diagnosis.<br>DDx contains something that is very close, but not an exact match to the probable diagnosis.<br>DDx includes the probable diagnosis. | Specialist |
| How close did the doctor's differential diagnosis (DDx) come to including any of the PLAUSIBLE ALTERNATIVE DIAGNOSES from the answer key? | 5-point scale | Nothing in the DDx is related to any of the plausible alternative diagnoses.<br>DDx contains something that is related, but unlikely to be helpful in determining any of the plausible alternative diagnoses.<br>DDx contains something that is closely related and might have been helpful in determining one of the plausible alternative diagnoses.<br>DDx contains something that is very close, but not an exact match to any of the plausible alternative diagnoses.<br>DDx includes at least one of the plausible alternative diagnoses. | Specialist |
| **Management** | | | |
| Was the doctor's recommendation appropriate as to whether an escalation to a non-text consultation is needed, e.g. video or in-person (without which an appropriate investigation/management plan cannot be decided)? | 4-point scale | No - Escalation was required but not performed. Failure to escalate to video or in-person assessment could have caused harm<br>No - Escalation was performed unnecessarily<br>Yes - Escalation was required and performed<br>Yes - Escalation was not required and not performed | Specialist |
| Did the doctor SUGGEST appropriate INVESTIGATIONS? | 3-point scale | No - The doctor did not recommend investigations, but the correct action would be to order investigations<br>No - The doctor recommended investigations but these were not comprehensive (some were missing)<br>Yes - The doctor recommended a comprehensive and appropriate set of investigations (including correctly selecting zero investigations if this was best for the case) | Specialist |
| Did the doctor AVOID INappropriate INVESTIGATIONS? | Binary scale | Yes<br>No | Specialist |
| Did the doctor SUGGEST appropriate TREATMENTS? | 3-point scale | No - The doctor did not recommend treatments, but the correct action would be to recommend investigations<br>No - The doctor recommended treatments but these were not comprehensive (some were missing)<br>Yes - The doctor recommended a comprehensive and appropriate set of treatments (including correctly selecting zero treatments if this was best for the case or if further investigation should precede treatment) | Specialist |
| Did the doctor AVOID INappropriate TREATMENTS? | Binary scale | Yes<br>No | Specialist |
| To what extent was the doctor's MANAGEMENT PLAN appropriate, including recommending emergency or red-flag presentations to go to ED? | 5-point scale | Very Inappropriate<br>Inappropriate<br>Neither Appropriate Nor Inappropriate<br>Appropriate<br>Very Appropriate | Specialist |
| Was the doctor's recommendation about a FOLLOW-UP appropriate? | 4-point scale | No - A follow-up was needed but the doctor failed to mention this<br>No - A follow-up was not needed but the doctor unnecessarily suggested one<br>Yes - A follow-up was needed and the doctor recommended an appropriate follow-up<br>Yes - A follow-up was not needed and the doctor did not suggest it | Specialist |
| **Confabulation** | | | |
| Did the doctor CONFABULATE anything, either within the consultation or in their responses to the post-questionnaire? | Binary scale | Yes, there are confabulations<br>No confabulations | Specialist |

Tao Tu, Mike Schakermann,
Alan Karthikesalingam,
Vivek Natarajan

# Reporting Summary

Please do not complete any field with "not applicable" or n/a.  Refer to the help text for what text to use if an item is not relevant to your study.
For final submission: please carefully check your responses for accuracy; you will not be able to make changes later.

## Statistics

For all statistical analyses, confirm that the following items are present in the figure legend, table legend, main text, or Methods section.

| n/a | Confirmed | |
|---|---|---|
| ☐ | ☒ | The exact sample size ($n$) for each experimental group/condition, given as a discrete number and unit of measurement |
| ☐ | ☒ | A statement on whether measurements were taken from distinct samples or whether the same sample was measured repeatedly |
| ☐ | ☒ | The statistical test(s) used AND whether they are one- or two-sided <br> *Only common tests should be described solely by name; describe more complex techniques in the Methods section.* |
| ☐ | ☒ | A description of all covariates tested |
| ☐ | ☒ | A description of any assumptions or corrections, such as tests of normality and adjustment for multiple comparisons |
| ☐ | ☒ | A full description of the statistical parameters including central tendency (e.g. means) or other basic estimates (e.g. regression coefficient) AND variation (e.g. standard deviation) or associated estimates of uncertainty (e.g. confidence intervals) |
| ☐ | ☒ | For null hypothesis testing, the test statistic (e.g. $F$, $t$, $r$) with confidence intervals, effect sizes, degrees of freedom and $P$ value noted <br> *Give P values as exact values whenever suitable.* |
| ☒ | ☐ | For Bayesian analysis, information on the choice of priors and Markov chain Monte Carlo settings |
| ☒ | ☐ | For hierarchical and complex designs, identification of the appropriate level for tests and full reporting of outcomes |
| ☒ | ☐ | Estimates of effect sizes (e.g. Cohen's $d$, Pearson's $r$), indicating how they were calculated |

*Our web collection on statistics for biologists contains articles on many of the points above.*

## Software and code

Policy information about availability of computer code

| Data collection | The algorithms and scripts were implemented using Python 2.7.18 for data collection |
|---|---|
| Data analysis | The data analysis scripts were implemented in Python 2.7.18. We will not be able  to open source the LLMs used in this study. |

Policy information about availability of data

All manuscripts must include a data availability statement. This statement should provide the following information, where applicable:
- Accession codes, unique identifiers, or web links for publicly available datasets
- A description of any restrictions on data availability
- For clinical datasets or third party data, please ensure that the statement adheres to our policy

We used the opensource MedQA dataset for training of the models. We also used openly available scenario packs from the UK MCR website for the conduct of the OSCE study. -
https://www.thefederation.uk/sites/default/files/documents/Station%202%20Scenario%20Pack%20%2816%29.pdf

# Research involving human participants, their data, or biological material

Policy information about studies with human participants or human data. See also policy information about sex, gender (identity/presentation), and sexual orientation and race, ethnicity and racism.

| | |
|---|---|
| Reporting on sex and gender | n/a |
| Reporting on race, ethnicity, or other socially relevant groupings | n/a |
| Population characteristics | n/a |
| Recruitment | n/a |
| Ethics oversight | n/a |

Note that full information on the approval of the study protocol must also be provided in the manuscript.

# Field-specific reporting

Please select the one below that is the best fit for your research. If you are not sure, read the appropriate sections before making your selection.

[X] Life sciences     [ ] Behavioural & social sciences     [ ] Ecological, evolutionary & environmental sciences

For a reference copy of the document with all sections, see nature.com/documents/nr-reporting-summary-flat.pdf

# Life sciences study design

All studies must disclose on these points even when the disclosure is negative.

| | |
|---|---|
| Sample size | The study involved 159 OSCE scenarios. No sample size calculation was performed. |
| Data exclusions | No data was excluded |
| Replication | The evaluations in the study were performed by multiple specialist physicans and patient actors. Triplicate ratings were obtained from specialists. The actors were sourced from two different institutions in two separate countries and the OSCE scenario packs were from three countries. The results of the study were consistent across all of them. |
| Randomization | The order in which (a) patient actors completed both study arms, and (b) specialists assessed quality for both study arms, was randomized. |
| Blinding | Patient actors and specialist evaluators were not told which study arm they were exposed to during text-based conversation and evaluation respectively. |

# Behavioural & social sciences study design

All studies must disclose on these points even when the disclosure is negative.

| | |
|---|---|
| Study description | |
| Research sample | |
| Sampling strategy | |
| Data collection | |
| Timing | |
| Data exclusions | |
| Non-participation | |
| Randomization | |

# Ecological, evolutionary & environmental sciences study design

All studies must disclose on these points even when the disclosure is negative.

| | |
|---|---|
| Study description | |
| Research sample | |
| Sampling strategy | |
| Data collection | |
| Timing and spatial scale | |
| Data exclusions | |
| Reproducibility | |
| Randomization | |
| Blinding | |

Did the study involve field work?  ☐ Yes  ☐ No

## Field work, collection and transport

| | |
|---|---|
| Field conditions | |
| Location | |
| Access & import/export | |
| Disturbance | |

# Reporting for specific materials, systems and methods

We require information from authors about some types of materials, experimental systems and methods used in many studies. Here, indicate whether each material, system or method listed is relevant to your study. If you are not sure if a list item applies to your research, read the appropriate section before selecting a response.

## Materials & experimental systems

| n/a | Involved in the study |
|---|---|
| ☒ | ☐ Antibodies |
| ☒ | ☐ Eukaryotic cell lines |
| ☒ | ☐ Palaeontology and archaeology |
| ☒ | ☐ Animals and other organisms |
| ☒ | ☐ Clinical data |
| ☒ | ☐ Dual use research of concern |
| ☒ | ☐ Plants |

## Methods

| n/a | Involved in the study |
|---|---|
| ☒ | ☐ ChIP-seq |
| ☒ | ☐ Flow cytometry |
| ☒ | ☐ MRI-based neuroimaging |

## Antibodies

| | |
|---|---|
| Antibodies used | |
| Validation | |

## Eukaryotic cell lines

Policy information about cell lines and Sex and Gender in Research

| | |
|---|---|
| Cell line source(s) | |
| Authentication | |
| Mycoplasma contamination | |
| Commonly misidentified lines<br>(See ICLAC register) | |

## Palaeontology and Archaeology

| | |
|---|---|
| Specimen provenance | |
| Specimen deposition | |
| Dating methods | |

☐ Tick this box to confirm that the raw and calibrated dates are available in the paper or in Supplementary Information.

| | |
|---|---|
| Ethics oversight | |

Note that full information on the approval of the study protocol must also be provided in the manuscript.

## Animals and other research organisms

Policy information about studies involving animals; ARRIVE guidelines recommended for reporting animal research, and Sex and Gender in Research

| | |
|---|---|
| Laboratory animals | |
| Wild animals | |
| Reporting on sex | |
| Field-collected samples | |
| Ethics oversight | |

Note that full information on the approval of the study protocol must also be provided in the manuscript.

## Clinical data

Policy information about clinical studies

All manuscripts should comply with the ICMJE guidelines for publication of clinical research and a completed CONSORT checklist must be included with all submissions.

| | |
|---|---|
| Clinical trial registration | |
| Study protocol | |
| Data collection | |
| Outcomes | |

## Dual use research of concern

Policy information about dual use research of concern

### Hazards

Could the accidental, deliberate or reckless misuse of agents or technologies generated in the work, or the application of information presented in the manuscript, pose a threat to:

| No | Yes | |
|---|---|---|
| ☐ | ☐ | Public health |
| ☐ | ☐ | National security |
| ☐ | ☐ | Crops and/or livestock |
| ☐ | ☐ | Ecosystems |
| ☐ | ☐ | Any other significant area |

## Experiments of concern

Does the work involve any of these experiments of concern:

| No | Yes | |
|---|---|---|
| ☐ | ☐ | Demonstrate how to render a vaccine ineffective |
| ☐ | ☐ | Confer resistance to therapeutically useful antibiotics or antiviral agents |
| ☐ | ☐ | Enhance the virulence of a pathogen or render a nonpathogen virulent |
| ☐ | ☐ | Increase transmissibility of a pathogen |
| ☐ | ☐ | Alter the host range of a pathogen |
| ☐ | ☐ | Enable evasion of diagnostic/detection modalities |
| ☐ | ☐ | Enable the weaponization of a biological agent or toxin |
| ☐ | ☐ | Any other potentially harmful combination of experiments and agents |

# Plants

| Seed stocks | |
|---|---|
| Novel plant genotypes | |
| Authentication | |

# ChIP-seq

## Data deposition

☐ Confirm that both raw and final processed data have been deposited in a public database such as GEO.

☐ Confirm that you have deposited or provided access to graph files (e.g. BED files) for the called peaks.

| Data access links
*May remain private before publication.* | |
|---|---|
| Files in database submission | |
| Genome browser session
(e.g. UCSC) | |

## Methodology

| Replicates | |
|---|---|
| Sequencing depth | |
| Antibodies | |
| Peak calling parameters | |
| Data quality | |

Software

# Flow Cytometry

## Plots

Confirm that:

☐ The axis labels state the marker and fluorochrome used (e.g. CD4-FITC).

☐ The axis scales are clearly visible. Include numbers along axes only for bottom left plot of group (a 'group' is an analysis of identical markers).

☐ All plots are contour plots with outliers or pseudocolor plots.

☐ A numerical value for number of cells or percentage (with statistics) is provided.

## Methodology

Sample preparation

Instrument

Software

Cell population abundance

Gating strategy

☐ Tick this box to confirm that a figure exemplifying the gating strategy is provided in the Supplementary Information.

# Magnetic resonance imaging

## Experimental design

Design type

Design specifications

Behavioral performance measures

Imaging type(s)

Field strength

Sequence & imaging parameters

Area of acquisition

Diffusion MRI          ☐ Used          ☐ Not used

## Preprocessing

Preprocessing software

Normalization

Normalization template

Noise and artifact removal

Volume censoring

## Statistical modeling & inference

Model type and settings

Effect(s) tested

Specify type of analysis: ☐ Whole brain ☐ ROI-based ☐ Both

Statistic type for inference

(See Eklund et al. 2016)

Correction

## Models & analysis

| n/a | Involved in the study |
|-----|------------------------|
| ☐ | ☐ Functional and/or effective connectivity |
| ☐ | ☐ Graph analysis |
| ☐ | ☐ Multivariate modeling or predictive analysis |

Functional and/or effective connectivity

Graph analysis

Multivariate modeling and predictive analysis

