## [Peer Review File · Nature]

Towards Conversational Diagnostic AI

Corresponding Author: Dr Tao Tu

Version 1:

Reviewer comments:

Referee #2

(Remarks to the Author)

This is a very interesting paper by (roughly) the group that has explored and published on building large language model based systems that perform at human or better levels on clinical diagnostic challenges. Here, they extend that work to include dialogue with (simulated) patients to take patient histories to use with their diagnostic system to suggest appropriate diagnoses based on the information they have gathered through dialog with these patient stand-ins. They demonstrate that their system achieves consistently higher top-k diagnostic performance on case scenarios than primary care providers given the same tasks. In addition, using standards developed to evaluate the performance of human doctors, their system was judged superior (and never significantly lower) on various communication characteristics such as empathy.

Interestingly, the superior performance of their AMIE system appears rooted in its diagnostic ability, not its history-gathering. A nice ablation study shows that the diagnostic engine achieves very similar diagnostic results from its own gathered information as from the PCP-gathered data, suggesting that it is not information gathering that is the root of the better performance.

From my reading, the key to AMIE's excellent performance comes from a "self-play" training augmentation inspired by systems such as Alpha-Go, though such a system is far more difficult to implement in a medical diagnosis task. In simulated Go games, the outcome is easily evaluated, since there are definitive criteria for winning or losing. In the diagnostic "game", some goals such as top-k performance can be assessed for the ~150 case scenarios they have, but not as reliably for the myriad generated cases used in this self training setup. In fact, for me the most remarkable aspect of this paper is that this inner/outer self-play loop, which relies on its own language model for not only playing the roles of patient, doctor, moderator and critic, but also uses its own evolving LLM to perform the internal evaluations needed in this training. My expectation was that such a system, relying on its own capabilities to help train itself better, would likely "go off the rails" and become worse rather than better, but this appears not to happen. I was also surprised that the web search based vignette generator did not lead to the kinds of extreme demographic biases reported, for example, in Zack T, et al. Assessing the potential of GPT-4 to perpetuate racial and gender biases in health care: a model evaluation study. *Lancet Digital Health* 2024 Jan;6:e12–22. I was also struck by the dramatically more concise

conversations in the simulated data than in actual transcribed conversations, which might be another source of misleading training. I would welcome some additional discussion of these aspects of AMIE's organization and use to help readers better understand why it works, not just that it works well.

I appreciated the cautions and limitations expressed in the paper, and the thorough range of evaluations of their system. Especially important, I think, would be a further exploration of how a system such as this could actually be useful in clinical practice. The paper does mention deferral to human experts, but exploring such ideas would require a large number of additional capabilities not present in the current system, such as assessing in real-time the quality of actions taken by a human PCP. I also suspect that, except in very limited circumstances, a turn-based text conversation with either a human expert or an AI-based one would not be a natural mode of patient communication. Although the COVID pandemic has made tele-consultations more common than before, these typically include speech rather than typing as a communication means and video that can allow for more demonstrative communication, such as showing a skin lesion or demonstrating a range of motion constraint, not to speak of the importance of non-verbal communication of patients' affect, pain, etc.

I encountered a few other, more minor issues:

I found the discussion of instruction fine tuning somewhat mysterious.

I could not see why the entire system is built on PaLM 2, but the auto-evaluation of differential diagnoses was based on Med-PaLM 2. In what significant ways does the training of Med-PaLM 2 differ from the training of PaLM 2 to create AMIE?

(Remarks on code availability)

Referee #3

(Remarks to the Author)

This study represents a major milestone in health communication science. Congratulations. It is highly original and potentially transformative and, with further elaboration, will hold major significance in the field. The datasets and associated methods to develop the tool were well laid out, although the numerous references of pre-published studies means the authors will need to expand on the methods section to fill in the gaps. The authors often place results in inappropriate sections--an issue that will need to be addressed. The same can be said for some introductory sections. The discussion section needs to be enriched with a deeper discussion of its limitations related to clinical epidemiology and Bayesian approaches to diagnosis. I provide detailed critiques and recommendations below:

Introduction

Line 17. Authors should acknowledge that there actually is wide variation in physicians' communication skills, not just that they all have this expertise.

Lines 27-28 re reducing disparities should be seriously qualified, as most health tech advances have been shown to widen health disparities. Work in the development, implementation and policy stages would need to be done for this to happen.

Line 33 re "expert clinicians" should be changed, as this study did not assess AI performance vs expert clinicians, but just practicing generalist physicians who were board-certified. I would use the term "practicing generalist physicians."

Lines 42-43. Expand domains to include communication efficacy (or conversation quality).

Line 55. What does "counterbalanced" mean in this context?

Lines 44-56. I am not sure if Nature allows for a summarization of results in an Introduction.

Section 2.1

Figures 1 and 2 are not called out in the text until too late.

Line 77 while MedQA questions are appropriate for people with known disease (where the question is which one does the patient have?), the approach does not capture one of the most important roles of a generalist physician: to distinguish symptoms that represent true disease states from non-disease states. While the real-world dialogue dataset may fill this gap, there is no clarity in the Methods as to how/if this was achieved. More on this limitation later.

Line 80 It is critical to know whether any of the 191 "curated" MedQA questions/clinical reasoning items had overlap with

those tested in the RCT. This has implications.

Relatedly, nearly all of the references related to the Methods in section 2 are pre-publications. This makes it very challenging to evaluate the study's developmental methods. The journal will need to decide how it wants to handle this issue in a field that is so rapidly evolving. At a minimum, I suggest the authors provide more details in all of section 2, and other sections that lack peer-reviewed sources.

Line 84. I fail to understand how, for medical summarization, a dataset that contains 2 million ICU notes would be helpful for outpatient, clinic-based history-taking, diagnostic reasoning, and conversational quality. If anything, it would be counterproductive.

Section 2.2

Line 195. I understand how the self-critic was used to assess concordance with ground truth around diagnosis, as the diagnosis was already established at the time of the vignette setting. But how were the other "most likely differential diagnoses" ascertained?

Line 202 It is unclear to this reviewer how fine tuning is different from the 4 actor process described in the prior section—or what specific challenge it was being used to overcome. Also, clearer language re what was done in fine tuning is requested, without having to read a technical report.

Line 240. Why does step 3 come after Step 2? Order seems out of whack.

Section 3.

Lines 257-259. The description of "pilot work" with "clinicians" and scale development is scanty at best. This needs significant work to fill in the blanks, especially since these represented some of the main outcomes. In addition, it is unclear that "patient actors" are appropriate delegates for real patients when determining what is important to a patient.

Section 3.1 This reviewer notes that OSCEs are typically time-limited and clinicians are conditioned to limit their time. This has implications for the limitations section.

Section 3.2

Line 286. There should be a supplement that describes the scenarios in the 149 scenario packs (actually, just the ground truth diagnoses would suffice). In addition, the distribution of specialties from which scenarios were selected should be based on some epidemiologic truth around what generalist physicians are presented with in day to day practice.

Line 319. Significant bias is introduced by engaging specialist physicians to be the expert/external raters rather than generalists. Specialists see only a very small and selected percentage of patients than that of PCPs, and therefore their patients have much different pre-test and post-test probabilities of true disease. In addition, it is widely recognized that generalists tend to have more "patient-centered" communication skills and are more likely to be experienced in rating learners on these skills than specialist are, again making this choice puzzling. Greater justification is needed for making this choice, and one wonders whether this bias was also built in to the AI algorithms' treatment plans/recommendations (i.e. escalate vs test of time). It would be important to clarify whether in the OSCE agents' responses, they could suggest 'watch and wait' or other conservative measures, as some of the discrepancies observed have to do with epidemiologic differences in patient populations.

Line 322. No mention is made of the number of specialist physicians used for expert ratings, whether there was double coding, or whether there was any reliability testing or training performed. This is a major weakness for all expert rater outcomes.

Line 336. The "ideal" criteria for rating OSCEs that were provided to expert raters needs significant explication.

Section 3.3

Line 347-52. Authors need to define "good alignment" between CoT auto-evaluation and expert raters etc.

Section 3.4

Line 364. Does top k here mean that the diagnosis was one of the top 3 diagnoses based on the ground truth? Please define.

Section 4

Line 378. The fact that PCPs provided fewer diagnoses (mean 5) compared to AIME (mean 10) has implications for the analyses you present in Figure 3, as the number of PCPs does not stay at 20 as one moves down the x-axis.

Section 4.2

Line 417. Explanation of non-responses by patient actors is needed, as well as how these were handled statistically.

Section 5

This section seems like it should be in a Discussion section (Section 6). It does not present new results but frames the work presented.

Section 6

Line 508. The comment about "challenging cases" suggests that AIME may be most appropriate in specialist settings, not in primary care.

Lines 515, 518, 520. New results are presented. All results must be presented in the Results section.

Line 553. Replace with "spent"

Line 567. This endpoint is unrealistic in the real world as well and should be acknowledged.

Line 637. Re equity, add reference re physicians' variation in communication skills across health literacy. See <https://www.science.org/doi/10.1126/sciadv.abj2836>

Line 645. Define red teaming

Line 652. Choose alternate term to "low-resource languages". Add in issues of literacy.

Limitation section needs to be expanded. Two straightforward limitations have to do with (a) toning down the issue of comprehensiveness/completeness. Few clinical presentations to primary care require a list of differential diagnoses that are 10 diagnoses long, and AMIE was programmed to generate a list of 10; and (b) expanding on the non-representativeness of patient actors and the entire experiment especially as it relates to patient's language and literacy limitations—including their difficulties in composing and reading messages, as well as difficulties conveying clear histories and responding to clinical questions in "logical" ways (from a medical perspective).

But the greatest limitation is that AIME appears to have been designed and tested under an assumption that all patients have pathology. This suggests that the creators were not aware of the population-level epidemiologic realities of primary care, where well over 80% of the work in assessing novel symptoms and syndromes has to do with ruling out disease, not ruling it in. Imagine a scenario like the one the authors raise in the discussion section: one in which users (not presenting to clinical settings) use AIME like Chat GPT—engaging in diagnostic conversations with an algorithmic conversational agent whose job is to make diagnoses. The results could be disastrous—generating a population that believes it has serious pathology, the kind that demand testing and (perhaps) unnecessary utilization of services. Making a diagnosis is only a small part of clinical reasoning. This problem is exacerbated by the choice the authors made to engage specialists (expert in pathology, who care for a small portion of the populace that indeed is ill) to assess the performance of generalists (experts in health, who care for a large portion of the populace that indeed is not ill). These conflicting roles need to be addressed and reconciled in the limitations section.

Many Figures (A.9, A15) may need to be presented as Supplemental Tables or information. Right now, much is not formatted for a scientific journal such as Nature. The journal editors will need to make specific recommendations based on their standards.

(Remarks on code availability)

Code was not provided.

Version 2:

Reviewer comments:

Referee #2

(Remarks to the Author)

Overall, I am impressed that the authors have responded very well to both my critique of the original submission and the comments from Reviewer #3. I had felt that the work reported here is quite innovative, important, and carefully done, so it deserves a high profile publication. The additions spurred by our critiques make it even better.

Most of my residual issues in this second review are just editing questions:

Fig 1: overlap of "Outer Self-play" with "Patient's Confidence"

356-360 repeats description of the randomization of order in which a patient actor interacts with a PCP or AMIE.

Figs A.7 and A.8: The captions duplicate the description of the (a) graphs and (b) graphs, whereas obviously the two report different results.

It occurred to me that the authors seem to have missed an opportunity, in the spirit of Turing, to ask patient actors to identify whether they thought they were conversing with a PCP or AMIE. The result might shed further light on their discussion of the artificiality of the communication scenario and the observed differences between judgments of PCPs and AMIE.

--Peter Szolovits

(Remarks on code availability)

Referee #3

(Remarks to the Author)

The authors did a remarkable job addressing this reviewer's concerns and I believe that the revised manuscript should be accepted under two contingencies:

1. The exploratory findings re the poor performance of AIME under the scenario in which the patient has limited English

skills should be included in this paper (not just a response to a reviewer), as it has terribly important relevance to the earlier discussion re health equity. Including these results does not take away from the transformative nature of this work; rather, it points to the intent of the team to continue to refine the tool to ensure that its benefits do not only accrue to those with the least needs for such innovation. If desired, the authors could limit the results to just those with limited English proficiency, as opposed to the other "personality types."

2. The necessary changes in formatting and placement of sections will need to be carefully reviewed by the editor, as for the most part, the authors did not address these at this point in their revise and resubmit.

(Remarks on code availability)

As I had mentioned when I was invited to review this work, I am not expert in LLM coding.

Version 3:

Reviewer comments:

Referee #2

(Remarks to the Author)

I have previously reviewed this article and recommended publication. I believe the current version differs only in response to some editorial requests, and I am still pleased with it.

(Remarks on code availability)

Response to Referee #2

Comment: This is a very interesting paper by (roughly) the group that has explored and published on building large language model based systems that perform at human or better levels on clinical diagnostic challenges. Here, they extend that work to include dialogue with (simulated) patients to take patient histories to use with their diagnostic system to suggest appropriate diagnoses based on the information they have gathered through dialog with these patient stand-ins. They demonstrate that their system achieves consistently higher top-k diagnostic performance on case scenarios than primary care providers given the same tasks. In addition, using standards developed to evaluate the performance of human doctors, their system was judged superior (and never significantly lower) on various communication characteristics such as empathy.

Interestingly, the superior performance of their AMIE system appears rooted in its diagnostic ability, not its history-gathering. A nice ablation study shows that the diagnostic engine achieves very similar diagnostic results from its own gathered information as from the PCP-gathered data, suggesting that it is not information gathering that is the root of the better performance.

From my reading, the key to AMIE's excellent performance comes from a "self-play" training augmentation inspired by systems such as Alpha-Go, though such a system is far more difficult to implement in a medical diagnosis task. In simulated Go games, the outcome is easily evaluated, since there are definitive criteria for winning or losing. In the diagnostic "game", some goals such as top-k performance can be assessed for the ~150 case scenarios they have, but not as reliably for the myriad generated cases used in this self training setup.

In fact, for me the most remarkable aspect of this paper is that this inner/outer self-play loop, which relies on its own language model for not only playing the roles of patient, doctor, moderator and critic, but also uses its own evolving LLM to perform the internal evaluations needed in this training. My expectation was that such a system, relying on its own capabilities to help train itself better, would likely "go off the rails" and become worse rather than better, but this appears not to happen.

Response: We appreciate the reviewer's detailed and thoughtful review, and are happy to provide additional explanation regarding the setup of the self-play and intuition for why it improved performance without going "off the rails".

The reviewer notes that in our dialogue self-play setup, some "goals" (or what constitutes a high quality medical conversation) are hard to evaluate compared to well-defined criteria such as winning in simulated Go games. We have expanded our Discussion section on "Simulated Dialogue" as follows to explain this better accounting for the nuances:

"Additionally, the task of producing reward signals for the quality of medical diagnostic conversations is more challenging than evaluating outcomes in rule-based constrained environments where success is well-defined (e.g. winning or losing a game of Go). Our process

for generating synthetic vignettes was designed with this consideration in mind. Since we know the ground truth condition for each vignette and the corresponding simulated dialogue(s) rollout, we are able to automatically assess the correctness of AMIE's DDx predictions as a proxy reward signal. This reward signal is used to filter out 'unsuccessful' simulated dialogues, i.e., those for which AMIE fails to produce an accurate DDx prediction during this self-play process. Beyond DDx accuracy, the self-play critic agent also assessed other qualities including the level of empathy, professionalism and coherence conveyed by the doctor agent for each simulated dialogue. While these latter constructs are more subjective compared to diagnostic accuracy, they served as domain-specific heuristics imposed by clinical experts from our research team to help steer AMIE's development towards alignment with established clinical values. We also note that in our preliminary analysis described in this work, our auto-evaluation framework for assessing the conversations along such rubrics was found to be in good alignment with human ratings and comparable to the inter-specialist agreement on these criteria."

Comment: I was also surprised that the web search based vignette generator did not lead to the kinds of extreme demographic biases reported, for example, in Zack T, et al. Assessing the potential of GPT-4 to perpetuate racial and gender biases in health care: a model evaluation study. Lancet Digital Health 2024 Jan;6:e12–22.

Response: We appreciate the reviewer's thoughtful consideration regarding potential risks of bias amplification with our web search based vignette generation approach. Our goal of using web search as a mechanism of grounding for AMIE was to actually help mitigate the risk of potential demographic biases as reported in Zack T, et al. In Zack T, et al, the authors only used simple prompting of GPT4, which relies entirely on the model's internal memory and therefore is more susceptible to producing biased results. We used search to fetch a wide range of diverse but plausible demographics possible for each medical condition (a proxy to giving model access to "ground-truth" data similar to the true US disease prevalence estimates used by Zack T, et al). Specifically, for each condition, we retrieved 60 passages from an internet search engine to cover a diverse set of demographics, symptoms and management plans (followed by a relevance filtering step). The exact search query (for 1) and prompt template (for 2) are provided on page 7 of the manuscript. We further note that the data generated using the web-search vignettes are additionally filtered using our auto-evaluation framework before use in the model training. We have added the following clarification to Discussion section "Fairness and Bias":

"To mitigate the risk that our synthetic vignettes would skew towards certain demographic groups, we leveraged web search to retrieve a range of demographics and associated symptoms relevant to each condition. We used these as input to the prompt template for vignette generation, instructing the model to produce multiple different vignettes given this range of inputs. While this mechanism was designed with the intent of mitigating risks of bias amplification, a comprehensive evaluation of conversational diagnostic models like AMIE for equity, fairness and bias is an important scope for future work."

Comment: I was also struck by the dramatically more concise conversations in the simulated data than in actual transcribed conversations, which might be another source of misleading training. I would welcome some additional discussion of these aspects of AMIE's organization and use to help readers better understand why it works, not just that it works well.

Response: Referee #2 makes an excellent point regarding the fact that simulated dialogues from self-play had substantially fewer conversational turns than transcribed conversations from real-world dialogues. From a model training perspective, the model was trained to predict a particular turn (e.g. doctor turn) in the dialogue given all previous turns between the patient and doctor. Having training dialogues with variable length and style helps the model to generate the desired response under different levels of uncertainty (the longer the previous context, the more information is likely available for the model to make a prediction). Additionally, the average number of turns in OSCE dialogues (between human physicians and patient actors) is 20, consistent with our synthetic dialogues but we note that the sessions were instructed to be completed in twenty minutes.

We've added the following elaboration in Section 2.2 to contextualize this discrepancy:

"We note that the simulated dialogues from self-play had significantly fewer conversational turns than those from real-world data described in the previous section. This difference is expected given that our self-play mechanism was designed - via instructions to doctor and moderator agents - to simulate text-based conversations. By contrast, real-world dialogue data was transcribed from in-person encounters. There are fundamental differences in communication styles between text-based and face-to-face conversations. For example, in-person encounters may afford a higher communication bandwidth including a higher total word count and more 'back and forth' (i.e., greater number of conversational turns) between the physician and the patient. AMIE, by contrast, was designed for focused information gathering via a text chat interface."

Comment: I appreciated the cautions and limitations expressed in the paper, and the thorough range of evaluations of their system. Especially important, I think, would be a further exploration of how a system such as this could actually be useful in clinical practice. The paper does mention deferral to human experts, but exploring such ideas would require a large number of additional capabilities not present in the current system, such as assessing in real-time the quality of actions taken by a human PCP. I also suspect that, except in very limited circumstances, a turn-based text conversation with either a human expert or an AI-based one would not be a natural mode of patient communication. Although the COVID pandemic has made tele-consultations more common than before, these typically include speech rather than typing as a communication means and video that can allow for more demonstrative communication, such as showing a skin lesion or demonstrating a range of motion constraint, not to speak of the importance of non-verbal communication of patients' affect, pain, etc.

Response: This is an excellent point that warrants further research that was beyond the scope of our work. Our study was limited to synchronous text chat because it allowed for the

randomized, blinded study design and is what is most natural for LLMs like AMIE. Further research is needed to carefully develop multimodal AI systems capable of diagnostic dialogue and investigate how they could be used in real-world settings, where verbal and visual cues would be highly informative. Finally, we also need rigorous studies to understand the downstream logistics and evaluate how these systems influence clinical management and ultimately, patient outcomes.

Comment: I encountered a few other, more minor issues:
I found the discussion of instruction fine tuning somewhat mysterious.

Response: We have edited section 2.3 to add more clarity and detail regarding the fine-tuning procedure used to train AMIE.

*“For dialogue generation tasks, AMIE was **instructed to assume either the doctor or patient role, and given the dialogue up to a certain turn, to predict the next conversational turn.** When playing the patient agent, AMIE’s **instruction was to reply to the doctor agent’s questions about their symptoms, drawing upon information provided in patient scenarios. These scenarios included patient vignettes (see \cref{patient_vignettes}) for simulated dialogues or metadata such as demographics, visit reason, and diagnosis type for the real-world dialogue dataset. For each fine-tuning example in the patient role, the corresponding patient scenario was added to AMIE’s context.** In the doctor agent role, AMIE was **instructed to act as an empathetic clinician, interviewing patients about their medical history and symptoms to ultimately arrive at an accurate diagnosis. From each dialogue, we sampled on average 3 turns for each of the doctor and patient roles as the target turns to predict based on the conversation leading up to that target turn. Target turns were randomly sampled from all turns in the dialogue that had a minimum length of 30 characters.**”*

Comment: I could not see why the entire system is built on PaLM 2, but the auto-evaluation of differential diagnoses was based on Med-PaLM 2. In what significant ways does the training of Med-PaLM 2 differ from the training of PaLM 2 to create AMIE?

Response: We thank the reviewer for emphasizing this point of potential reader confusion. In our revised manuscript, we have added the following clarification in Section 3.3 “Auto-evaluation”:

“Note that both AMIE and Med-PaLM 2 share PaLM 2 as their base model. However, AMIE was fine-tuned on medical dialogue data and optimized for diagnostic conversations, Med-PaLM 2 was designed for the purpose of single-turn medical question answering, and not fine-tuned on medical dialogue data. As such, we considered Med-PaLM 2 sufficiently distinct from AMIE and, as a medical question answering engine, suited for the task of determining whether a predicted diagnosis (e.g. “COPD”) constituted a semantic match given a ground truth diagnosis (e.g., “Chronic Obstructive Pulmonary Disease”).”

Response to Referee #3

Comment: This study represents a major milestone in health communication science. Congratulations. It is highly original and potentially transformative and, with further elaboration, will hold major significance in the field. The datasets and associated methods to develop the tool were well laid out, although the numerous references of pre-published studies means the authors will need to expand on the methods section to fill in the gaps. The authors often place results in inappropriate sections--an issue that will need to be addressed. The same can be said for some introductory sections. The discussion section needs to be enriched with a deeper discussion of its limitations related to clinical epidemiology and Bayesian approaches to diagnosis. I provide detailed critiques and recommendations below:

Response: We appreciate the reviewer's thoughtful comments and suggestions, and are glad that the research was appreciated as original and potentially transformative. We provide a point-by-point response to individual comments below. We also agree that we could improve the organization of the work and will revisit the layout, formatting and organization of the manuscript with the journal editors prior to publication.

Comment: Introduction

Line 17. Authors should acknowledge that there actually is wide variation in physicians' communication skills, not just that they all have this expertise.

Response: We agree with this comment and have added the following sentence to Introduction (added passages in bold): ***"While there is wide variation in communication skills among clinicians, well-trained professionals can wield considerable skills in clinical history-taking and the wider "diagnostic dialogue"; however, access to this expertise remains episodic and globally scarce."***

Comment: Lines 27-28 re reducing disparities should be seriously qualified, as most health tech advances have been shown to widen health disparities. Work in the development, implementation and policy stages would need to be done for this to happen.

Response: This is indeed an important point to clarify, we thank the reviewer for raising this. We have incorporated the following elaboration in our revision (added passages in bold):

"The potential real-world utility of AI systems capable of clinical and diagnostic dialogue is broad, as the development of such capabilities might improve access to diagnostic and prognostic expertise, to improved quality, consistency, availability, and affordability of care. A health equity-centric approach to integrating such technology into existing workflows, which implies work in development, implementation, and policy stages, may have the potential to help realize better health outcomes (particularly for populations facing healthcare disparities)."

Comment: Line 33 re “expert clinicians” should be changed, as this study did not assess AI performance vs expert clinicians, but just practicing generalist physicians who were board-certified. I would use the term “practicing generalist physicians.”

Response: Thank you, we have changed this to “*practicing generalist physicians.*”

Comment: Lines 42-43. Expand domains to include communication efficacy (or conversation quality).

Response: We have added ‘*communication efficacy.*’

Comment: Line 55. What does “counterbalanced” mean in this context?

Response: Here “counterbalanced” means approximately in half of the cases, patient actors spoke to AMIE first and then to the PCP, and in the other half, they spoke to the PCP first and then to AMIE. This helps control for any potential order effects. We have added the following sentence (added sentence in BOLD) in the text to clarify:

*“The ordering of PCP and AMIE was randomized and patient actors were not informed as to which they were talking to in each consultation. **In half of the cases, patient actors interacted with the PCPs first, while in the other half of cases, patient actors interacted with AMIE first (counterbalanced design to control for any potential order effects).**”*

Comment: Lines 44-56. I am not sure if Nature allows for a summarization of results in an Introduction.

Response: Thank you to the reviewer for pointing this out, we will follow Nature guidelines once the work reaches the editorial stage (We believe Nature allows a separate section summarizing key contributions).

Comment: Section 2.1

Figures 1 and 2 are not called out in the text until too late.

Response: We moved up the reference to Figure 1 to the beginning of Section 2 and moved the reference to Figure 2 to the beginning of Section 3.

Comment: Line 77 while MedQA questions are appropriate for people with known disease (where the question is which one does the patient have?), the approach does not capture one of the most important roles of a generalist physician: to distinguish symptoms that represent true disease states from non-disease states. While the real-world dialogue dataset may fill this gap, there is no clarity in the Methods as to how/if this was achieved. More on this limitation later.

Response: We appreciate the reviewer's excellent point about the importance in the role of generalist physicians of distinguishing true disease states from non-disease states. We wholeheartedly agree with this consideration. We note that MedQA primarily tests for medical knowledge and is likely an example of a dataset that may help the model to learn how to distinguish between different states of disease rather than distinguish disease from non-disease. Per guidance from the reviewer, we have expanded our set of evaluation scenarios by **ten additional cases** representing a distribution of cases where there is no underlying disease state. This allows comparative/ablation analyses and we elaborate on this new data and the corresponding analyses in our response to the reviewer's last comment at the bottom of this document.

Comment: Line 80 It is critical to know whether any of the 191 "curated" MedQA questions/clinical reasoning items had overlap with those tested in the RCT. This has implications.

Response: We thank the reviewer for highlighting this possible issue. We can confirm that none of the clinical reasoning topics examined in curated MedQA questions were re-examined in the scenario packs created for our RCT-style evaluation.

Comment: Relatedly, nearly all of the references related to the Methods in section 2 are pre-publications. This makes it very challenging to evaluate the study's developmental methods. The journal will need to decide how it wants to handle this issue in a field that is so rapidly evolving. At a minimum, I suggest the authors provide more details in all of section 2, and other sections that lack peer-reviewed sources.

Response: We appreciate the reviewer's feedback. Our focus in this work is the development of a model optimized for diagnostic reasoning and conversations starting from a general purpose pre-trained LLM (i.e. PaLM 2). We believe each section has sufficient enough information for other interested researchers to replicate our methodology without using Google's PaLM 2 LLM (for example with open source model equivalents). We would like to point out that the development of the base LLM (PaLM 2) itself is not the focus of this work and is well known to be a highly complex and challenging endeavor and we refer our readers to the PaLM 2 technical report for details. We also additionally note that it is common practice in the ML community to cite preprints given the pace of research as the reviewer also points out. Finally, we have updated 12 references from preprints to the published versions as highlighted below and hope to do more such changes at the time of publication.

Below are all updated 12 references:

```
[12] @article{singhal2023large,  
  title={Large language models encode clinical knowledge},  
  author={Singhal, Karan and Azizi, Shekoofeh and Tu, Tao and Mahdavi, S Sara and Wei,  
Jason and Chung, Hyung Won and Scales, Nathan and Tanwani, Ajay and Cole-Lewis, Heather  
and Pfohl, Stephen and others},  
  journal={Nature},  
  volume={620},  
  number={7972},  
  pages={172--180},  
  year={2023},  
  publisher={Nature Publishing Group}  
}
```

```
[24] @inproceedings{sharma2021computational,  
  title={A Computational Approach to Understanding Empathy Expressed in Text-Based Mental  
Health Support},  
  author={Sharma, Ashish and Miner, Adam and Atkins, David and Althoff, Tim},  
  booktitle={EMNLP},  
  year={2021}  
}
```

```
[29] @inproceedings{he2022dialmed,  
  title={DialMed: A Dataset for Dialogue-based Medication Recommendation},  
  author={He, Zhenfeng and Han, Yuqiang and Ouyang, Zhenqiu and Gao, Wei and Chen,  
Hongxu and Xu, Guandong and Wu, Jian},  
  booktitle={Proceedings of the 29th International Conference on Computational Linguistics},  
  pages={721--733},  
  year={2022}  
}
```

```
[55] @article{ouyang2022training,  
  title={Training language models to follow instructions with human feedback},  
  author={Ouyang, Long and Wu, Jeffrey and Jiang, Xu and Almeida, Diogo and Wainwright,  
Carroll and Mishkin, Pamela and Zhang, Chong and Agarwal, Sandhini and Slama, Katarina  
and Ray, Alex and others},  
  journal={Advances in neural information processing systems},  
  volume={35},  
  pages={27730--27744},  
  year={2022}  
}
```

[56] @inproceedings{zhao2021ethical,
title={Ethical-Advice Taker: Do Language Models Understand Natural Language Interventions?},
author={Zhao, Jieyu and Khashabi, Daniel and Khot, Tushar and Sabharwal, Ashish and Chang, Kai-Wei},
booktitle={Findings of the Association for Computational Linguistics: ACL-IJCNLP 2021},
pages={4158--4164},
year={2021}
}

[62] @inproceedings{shor2023clinical,
title={Clinical BERTScore: An Improved Measure of Automatic Speech Recognition Performance in Clinical Settings},
author={Shor, Joel and Bi, Ruyue Agnes and Venugopalan, Subhashini and Ibara, Steven and Goldenberg, Roman and Rivlin, Ehud},
booktitle={Proceedings of the 5th Clinical Natural Language Processing Workshop},
pages={1--7},
year={2023}
}

[70] @inproceedings{varshney2022cdialog,
title={CDialog: A Multi-turn Covid-19 Conversation Dataset for Entity-Aware Dialog Generation},
author={Varshney, Deeksha and Zafar, Aizan and Behera, Niranshu and Ekbal, Asif},
booktitle={Proceedings of the 2022 Conference on Empirical Methods in Natural Language Processing},
pages={11373--11385},
year={2022}
}

[99] @article{gallegos2024bias,
title={Bias and fairness in large language models: A survey},
author={Gallegos, Isabel O and Rossi, Ryan A and Barrow, Joe and Tanjim, Md Mehrab and Kim, Sungchul and Deroncourt, Franck and Yu, Tong and Zhang, Ruiyi and Ahmed, Nesreen K},
journal={Computational Linguistics},
pages={1--79},
year={2024},
publisher={MIT Press 255 Main Street, 9th Floor, Cambridge, Massachusetts 02142, USA~...}
}

[103] @inproceedings{perez2022red,
title={Red Teaming Language Models with Language Models},

```
author={Perez, Ethan and Huang, Saffron and Song, Francis and Cai, Trevor and Ring, Roman and Aslanides, John and Glaese, Amelia and McAleese, Nat and Irving, Geoffrey},
booktitle={Proceedings of the 2022 Conference on Empirical Methods in Natural Language Processing},
pages={3419--3448},
year={2022}
}
```

```
[106] @inproceedings{ge2024mart,
title={MART: Improving LLM Safety with Multi-round Automatic Red-Teaming},
author={Ge, Suyu and Zhou, Chunting and Hou, Rui and Khabsa, Madian and Wang, Yi-Chia and Wang, Qifan and Han, Jiawei and Mao, Yuning},
booktitle={Proceedings of the 2024 Conference of the North American Chapter of the Association for Computational Linguistics: Human Language Technologies (Volume 1: Long Papers)},
pages={1927--1937},
year={2024}
}
```

```
[116] @inproceedings{ramesh2023fairness,
title={Fairness in Language Models Beyond English: Gaps and Challenges},
author={Ramesh, Krithika and Sitaram, Sunayana and Choudhury, Monojit},
booktitle={Findings of the Association for Computational Linguistics: EACL 2023},
pages={2106--2119},
year={2023}
}
```

```
[117] @inproceedings{hada2024large,
title={Are Large Language Model-based Evaluators the Solution to Scaling Up Multilingual Evaluation?},
author={Hada, Rishav and Gumma, Varun and Wynter, Adrian and Diddee, Harshita and Ahmed, Mohamed and Choudhury, Monojit and Bali, Kalika and Sitaram, Sunayana},
booktitle={Findings of the Association for Computational Linguistics: EACL 2024},
pages={1051--1070},
year={2024}
}
```

Comment: Line 84. I fail to understand how, for medical summarization, a dataset that contains 2 million ICU notes would be helpful for outpatient, clinic-based history-taking, diagnostic reasoning, and conversational quality. If anything, it would be counterproductive.

Response: We thank the reviewer for this comment. In practice, LLMs can benefit from training with a diverse set of tasks which prevents it from overfitting to one specific task and encourages

more robust behavior in overparameterized LLMs. To clarify, we added only 65 medical summarization examples curated from ICU notes to the training mixture (not 2 million). This helps the model to learn to pick out the most pertinent information from the ICU notes to produce summaries. It is likely in doing so it gets better at identifying pertinent clinical information in more general scenarios, including medical dialogue. Furthermore, the first step in our chain of reasoning process is to summarize patient information that has been gathered thus far, so learning how to summarize is directly useful for this. While this was beyond the scope of our current work, we also expect that in the future we will need future versions of AMIE to generate clinically useful summaries of the conversation interactions (SOAP notes) for oversight by clinicians.

Comment: Section 2.2

Line 195. I understand how the self-critic was used to assess concordance with ground truth around diagnosis, as the diagnosis was already established at the time of the vignette setting. But how were the other “most likely differential diagnoses” ascertained?

Response: This is a great question, as the vignettes for which these simulated dialogues were generated only contained a single ground truth diagnosis and not a DDX. The critic agent, given access to this single ground truth diagnosis, the full patient vignette (including the patient’s symptoms and demographics) and the dialogue between that patient and AMIE, was tasked with inferring other diagnoses and “matching” them with the doctor agent’s DDX. Given the high DDX accuracy, AMIE demonstrated in this and other works [1], we believe these inferred diagnoses are generally reasonable. We have added clarification to the text regarding this detail of the critic agent:

*“The doctor agent asks sufficient questions to identify at least two of the most likely differential diagnoses. They further refine their understanding through targeted questions towards the ground truth diagnosis and offer the corresponding treatment. **Note that because the patient vignettes themselves contained just one ground truth diagnosis, the critic had to infer whether any additional diagnoses discussed were reasonable for the given scenario.**”*

[1] McDuff, Daniel, et al. "Towards accurate differential diagnosis with large language models." *arXiv preprint arXiv:2312.00164* (2023)

Comment: Line 202 It is unclear to this reviewer how fine tuning is different from the 4 actor process described in the prior section—or what specific challenge it was being used to overcome. Also, clearer language re what was done in fine tuning is requested, without having to read a technical report.

Response: We appreciate this comment as the 4-actor process and the instruction fine-tuning process are highly complementary. Specifically, the 4-actor process is used to **generate** a large portion of the fine-tuning data (synthetic data), and the instruction fine-tuning process is used to

train the model to behave like the doctor/patient as encountered in this synthetically-generated fine-tuning data. We have added the following clarifications to the instruction fine-tuning section:

*“AMIE, built upon the base LLM PaLM 2~\cite{anil2023palm}, was instruction fine-tuned to enhance its capabilities for medical dialogue and reasoning. We refer to the PaLM-2 technical report for more details on the base LLM architecture. **Fine-tuning examples were crafted from the evolving simulated dialogue dataset generated by our four-agent procedure, as well as the static datasets. For each task, we designed task-specific instructions to instruct AMIE on what task it would be performing. For dialogue, this was assuming either the patient or doctor role within the conversation, while for the question-answering and summarization datasets, AMIE was instead instructed to answer medical questions or summarize EHR notes.** The first round of fine-tuning from the base LLM only used the static datasets, **while** subsequent rounds of fine-tuning leveraged the simulated dialogues generated through the self-play inner loop as described in \cref{Simulated data curation}.*

Comment: Line 240. Why does step 3 come after Step 2? Order seems out of whack.

Response: We thank the reviewer for the comment. To clarify, Step 1, analyzing patient information, is about synthesizing and summarizing the available information, and step 2, formulating the response, aims to draft a response based on this available information. However, it is possible for the model’s draft response from step 2 to have some mistakes, from minor issues in formatting to more major issues regarding factual inaccuracies and hallucinations. Step 3, refining the response, serves to edit and revise the drafted response to catch and fix some of these issues before sending the message out to the patient. This can be thought of as checking and self-verifying one's work before sending it to other examiners.

Comment: Section 3.

Lines 257-259. The description of "pilot work" with “clinicians” and scale development is scanty at best. This needs significant work to fill in the blanks, especially since these represented some of the main outcomes. In addition, it is unclear that “patient actors” are appropriate delegates for real patients when determining what is important to a patient.

Response: We thank the reviewer for this comment. As we have highlighted, our evaluation rubric is considerably more detailed than any prior approach we are aware of for testing the history-taking and conversational diagnostic abilities of an AI system. We have added more detail in the text at lines 257 onward to describe our scale development, alongside appropriate cautions that our evaluation rubric itself does not represent a validated “measurement instrument”.

“In contrast to these prior works we sought to anchor our human evaluation in criteria more commonly used for evaluating the quality of physicians’ expertise in history-taking, including their communication skills in consultation (see the three-step study design in Figure 2). Additionally, we aimed to evaluate conversation quality from the perspective of both the lay participant (the participating patient actor), and a non-participating professional observer (a physician that was not directly involved in the consultation).

We surveyed the literature and interviewed clinicians working as OSCE examiners in Canada and India to identify a minimum set of peer-reviewed published criteria that they considered comprehensively reflected the criteria that are commonly used in evaluating both patient-centered and professional-centered aspects of clinical diagnostic dialogue: identifying the consensus for best practices for patient-centered communication (PCCBP) in medical interviews, criteria examined for history-taking skills by the Royal College of Physicians in the UK as part of their Practical Assessment of Clinical Examination Skills (PACES), and criteria proposed by the UK General Medical Council Patient Questionnaire (GMCPQ) for doctors seeking patient feedback as part of professional re-validation.

In particular, PCCBP synthesizes prior literature to derive six consensus categories of function for the medical interview, representing best practice for patient-centered communication: fostering the relationship, gathering information, providing information, decision-making, enabling disease and treatment-related behavior, and responding to emotions. We converted each of these functions into a five-point Likert scale for evaluators to use in assessment. To help calibrate the judgment of evaluators applying Likert scores to each function, we shared the published behaviors given as indicators for how each function might be achieved. For example, in order to foster a relationship with the patient, the authors described that a physician can use appropriate language, show interest in the patient as a person, encourage patient participation, discuss mutual roles and responsibilities, express caring and commitment, engage in partnership building, appear open and honest, and build rapport and connection. We undertook a similar approach to converting PACES items into five-point Likert scales and deriving instructions for evaluators grounded in marksheets published by the Royal College of Physicians which provided evidence of each attribute. Within PACES and PCCBP, all categories were suited for evaluation from a professional-centered perspective, and only a subset of categories also lent themselves for evaluation from a patient-centered perspective. For example, the single category suitable for a patient-centered evaluation in PCCBP was relationship fostering, which allowed us to delve more deeply into this aspect from a patient perspective. We decided to collect more fine-grained annotations from patient actors regarding each of the individual criteria for relationship fostering (e.g. whether the physician used appropriate language, showed interest in the patient as a person, and so forth) to permit a deeper analysis, rather than applying a single five-point Likert scale. For GMCPQ, we implemented the published questionnaire format as is without further adaptation.

Inspired by prior approaches to evaluating LLM generations in clinical settings which noted that both appropriate and inappropriate generations can co-occur and that confabulation or hallucinations may be a risk specific to generative AI systems~\cite{singhal2023large}, we additionally evaluated the incidence of both appropriate and inappropriate investigation, management and follow-up plans; alongside looking for evidence of confabulation. As in our prior work, our overall framework items' form and wording were refined by undertaking pilot trials of the assessment and recording feedback from evaluating clinicians. Instructions for the clinicians were written including indicative examples of rating each item for a sample dialogue, and iterated until we observed convergence of the rating approaches by clinicians in the US, India and UK; to indicate our instructions for evaluation were usable. In our study, we conducted a rating of the physician perspective for all dialogues by a set of three independent physicians. For robustness, results were reported based on median rating score (or majority vote for binary ratings) among the three independent raters. Additionally, we provided evidence of inter-rater reliability for our framework in Section A.10.

The resulting evaluation framework enabled assessment from two perspectives: clinician and lay participants in the dialogues (i.e. patient actors). The framework included consideration of consultation quality, structure and completeness, the roles, responsibilities, and skills of the interviewer (Tables A.1, A.2, A.3, and A.4). We undertook a pragmatic approach to derive this usable framework and note that further development, extension and validation would be possible for a measurement instrument, including the application of instrument-construction approaches used elsewhere in social sciences~\cite{boateng2018best}”

We thank the reviewer for their comment that “it is unclear that “patient actors” are appropriate delegates for real patients when determining what is important to a patient.” We agree that this is an inherent limitation of all OSCE settings. Nonetheless, we believe that it is important to record both lay-participants’ and professional ratings of the dialogue, to reflect both aspects of quality. We have therefore added to the Discussion section (Evaluation Framework) as follows:

“We note that the lay participant perspective was represented in our study by patient actors (not real patients) and it is therefore unclear how accurately their perspectives would represent those of real patients in an actual clinical scenario (since the physical and mental lived experience of real disease symptoms and signs might alter a patients’ perspective of the encounter, in ways not replicable by a patient actor). However, as the raters were lay participants who themselves did partake in conversations with the physician (or AMIE), their subjective experience remains valuable to record and analyze, even if the extent to which it is a proxy for a real patient perspective cannot be assured. Furthermore, this possible limitation applies most saliently to evaluation criteria related to changes in the patient’s state of mind (e.g., GMCPQ Making Patient Feel At Ease) and less so to criteria regarding the doctor’s behavior (e.g., GCMPQ Being Polite), for which it could be expected that lived disease experience may have less impact.”

Comment: Section 3.1 This reviewer notes that OSCEs are typically time-limited and clinicians are conditioned to limit their time. This has implications for the limitations section.

Response: This is indeed correct. As we highlight in Section 3.2.1 patient actors were *“instructed to conclude the conversation after no more than 20 minutes”* to implement typical OSCE conventions. We appreciate the suggestion to elaborate on potential limitations regarding the time limit and have added the following sentence to the Discussion section (Conversational Performance):

“Note that the conversations in our study were time-limited to follow typical OSCE conventions. While real-world patient-physician consultations often also take place under time constraints, the specific time limit imposed in our study may not be reflective of real-world scenarios.”

Comment: Section 3.2

Line 286. There should be a supplement that describes the scenarios in the 149 scenario packs (actually, just the ground truth diagnoses would suffice). In addition, the distribution of specialties from which scenarios were selected should be based on some epidemiologic truth around what generalist physicians are presented with in day to day practice.

Response: We thank the reviewer for this suggestion, and have accordingly added to the appendix the list of ground truth conditions for each scenario pack (see Appendix Tables A.6 to A.12). Basing a selection on epidemiology for generalist physicians would have been a possible design choice, but would have required a considerably greater power and sample size if intended to provide some evaluative reflection of the scope and nature of “realistic” generalist primary care.

We discuss extensively in our paper that our experiment was not intended to provide a representative simulation of the experience of a doctor or patient in real primary care, as we consider this to be infeasible for an OSCE-type evaluation (it would also, for example, require multiple simulations of each disease presentation to try to reflect the within-condition variations in severity, timeliness, patient attributes and much more).

To select conditions for our OSCE, we provided indicative principles to our OSCE laboratory centers but did not pre-specify precise conditions in order to preserve the held-out nature of evaluation and mitigate risks of accidental model over-fitting to the test set during training or development. We provided the laboratories with links to a systematic review covering the most common conditions presenting to primary care, and recommended to focus primarily on scenarios in which it would be clinically reasonable to undertake an asynchronous text-based consultation (i.e. to consider to exclude those that would be impossible to conduct a clinical dialogue without a visual or in-person examination - <https://www.cfp.ca/content/64/11/832> as an example). As a second reference we suggested that the laboratories should adopt a curriculum-based approach to coverage by ensuring the scenario packs covered different

common physiologic systems (gastrointestinal, cardiac, respiratory, neurology, etc.) and, within these, diagnoses that reflected curriculum topics examined in generalist medicine by using local lists of common presenting complaints for curricula for competence in general medicine (as one open example:

<https://www.thefederation.uk/sites/default/files/Internal%20Medicine%20stage%201%20curriculum%20FINAL%20111217.pdf>).

Finally, we wanted to avoid interfering with and potentially contaminating our evaluation setup, hence beyond the general guidance, we relied on our OSCE laboratory centers' considerable expertise to guide the process.

Comment: Line 319. Significant bias is introduced by engaging specialist physicians to be the expert/external raters rather than generalists. Specialists see only a very small and selected percentage of patients than that of PCPs, and therefore their patients have much different pre-test and post-test probabilities of true disease. In addition, it is widely recognized that generalists tend to have more "patient-centered" communication skills and are more likely to be experienced in rating learners on these skills than specialists are, again making this choice puzzling. Greater justification is needed for making this choice, and one wonders whether this bias was also built into the AI algorithms' treatment plans/recommendations (i.e. escalate vs test of time). It would be important to clarify whether in the OSCE agents' responses, they could suggest 'watch and wait' or other conservative measures, as some of the discrepancies observed have to do with epidemiologic differences in patient populations.

Response: We thank the reviewer for this consideration. Regarding the reviewer's comment on evaluation of patient-centered communication - while we agree that theoretically the interpretation of a patient's perspective might differ between a generalist and a specialist (though we cannot find literature evidence to support that) - it is more likely that patient-actors themselves provide a better proxy for the experience of a patient in a conversation than their doctor counterpart. This is commonly how patient perspectives are provided in OSCE settings and patient-actors themselves, who had experience in performing a similar function for OSCE examination conduct in their institutions; but who also had undergone the first-party experience of participating in the simulated consultation in our study.

Regarding the reviewers' assertion that specialists see only a very small and selected series of patients; in our OSCE we sought to evaluate the quality of the differential diagnosis and management plan in a manner independent to the wider range of practice of an individual physician. We acknowledge that even for specific medical conditions, the optimal initial management (whether by generalist or specialist) is a matter of considerable debate (<https://jamanetwork.com/journals/jamainternalmedicine/fullarticle/411483>). Regarding the correct application of pre-test and post-test probability of disease; our study's measure of diagnostic accuracy (and the completeness and appropriateness of the differential diagnosis) was independent of whether the evaluator was a specialist or a generalist - it was determined by comparison to a scenario-pack specific "ground truth", which was itself determined by the OSCE

laboratories. The specialists rated the quality of the completeness and rigor of history-taking, and other MRCP-scale specific criteria, such as the clarity of explanations provided to the patient. We agree with the reviewer that specialists have a higher-case volume focused on specific conditions and their differentials. We therefore felt they were ideally suited to be the most stringent/harsh possible evaluation standard of our AI system's history-taking in specific scenarios; varying the specialist to be specific to each scenario. The same standard was applied both to PCP participants and the AI arm, and therefore did not introduce bias between groups.

Regarding the reviewer's question about the option to recommend a watch-and-wait strategy, OSCE agents (AI and PCP) were free to select "No, does not require escalation" in the post-questionnaire step (as an alternative to recommending escalation to either video-call or in-person consultation or both). During independent evaluation of OSCE agent responses, raters were asked to determine one of the following (per Table A.4 | Diagnosis and Management rubric details):

- No - Escalation was required but not performed. Failure to escalate to video or in-person assessment could have caused harm
- No - Escalation was performed unnecessarily
- Yes - Escalation was required and performed
- Yes - Escalation was not required and not performed

The evaluation rubric also included measurement of whether the OSCE agent selected an appropriate strategy for "Treatment", and for "Follow-up". This also appropriately encompassed the options of "conservative management" or "watch and wait", which participants were able to utilize in their normal manner.

Comment: Line 322. No mention is made of the number of specialist physicians used for expert ratings, whether there was double coding, or whether there was any reliability testing or training performed. This is a major weakness for all expert rater outcomes.

Response: We appreciate the reviewer's request for elaboration on specialist physician raters including replication, inter-rater reliability and rater training. Per our response to Reviewer 3's previous comment above, instructions for the clinicians were written including indicative examples of rating each item for a sample dialogue. Our original submission was based on ratings from just a single specialist physician per case. In response to Reviewer 3's excellent comment, we collected additional data and thus increased the replication to three independent ratings per case. These triplicate ratings allowed us to perform inter-rater reliability analysis which we detail in Appendix Section A.10 in the revised manuscript:

"A.10 Inter-rater Reliability Analysis for Specialist Physician Ratings."

We performed inter-rater reliability (IRR) analysis for specialist physician ratings. During the study, each case was independently rated by a set of three specialist physicians. In this setup, each specialist physician rated output from both AMIE and PCP for each case, in a randomized and blinded manner. Inter-rater agreement was measured as Randolph's κ [1] with respect to the binary outcome of whether the specialist physician rated AMIE at least as well as the PCP (or better). This corresponds to the primary outcome of interest for this work. Randolph's κ was more appropriate than other measures such as Krippendorff's α given the low baseline negative rate for several axes, i.e., in many cases AMIE was rated as on par with the PCP or better. We recruited separate sets of specialist raters to match each geographic location and medical specialty for a given scenario. To address the fact that different subsets of scenarios were rated by different sets of raters, κ values were first computed per location and specialty separately and then averaged. Table A.5 tabulates the resulting average κ values. Raters were in 'substantial' ($\kappa > 0.6$) or 'almost perfect' ($\kappa > 0.8$) agreement for 31 out of 32 evaluation axes, and in 'moderate' agreement for the remaining 'Eliciting Medication History' axis."

[1] Randolph, J. J. Free-Marginal Multirater Kappa (multirater K [free]): An Alternative to Fleiss' Fixed-Marginal Multirater Kappa. Online submission (2005).

Comment: Line 336. The "ideal" criteria for rating OSCEs that were provided to expert raters needs significant explication.

Response: We thank the reviewer for this suggestion. In the manuscript we stated that the application of our evaluation rubric was additionally informed by scenario-specific considerations:

"Each OSCE scenario pack additionally supplied specialists with scenario-specific clinical information to assist with rating the clinical quality of the consultation, such as the ideal investigation or management plans; or important aspects of the clinical history that would ideally have been elucidated for the highest quality of consultation possible."

To provide additional information regarding this, we have also now included a reference to fully-worked examples of the scenario-specific information, as this is common practice in OSCE examinations:

"Each OSCE scenario pack additionally supplied specialists with scenario-specific clinical information to assist with rating the clinical quality of the consultation, such as the ideal investigation or management plans; or important aspects of the clinical history that would ideally have been elucidated for the highest quality of consultation possible. ***This follows the common practice for instructions for OSCE examinations, in which specific clinical scenario-specific information is provided to ensure consistency amongst examiners, and follows the paradigm demonstrated by MRCP sample packs. For example, the scenario available at*** [***https://www.thefederation.uk/sites/default/files/Station%20%20Scenario%20Pack%20%20***](https://www.thefederation.uk/sites/default/files/Station%20%20Scenario%20Pack%20%20)

2816%29.pdf) informs an examiner that for a scenario in which the patient-actor has haemoptysis, the appropriate investigations would include chest X-ray, high-resolution CT scan of chest, bronchoscopy, spirometry; while bronchiectasis treatment options a candidate should be aware of should include chest physiotherapy, mucolytics, bronchodilators, and antibiotics.”

Comment: Section 3.3

Line 347-52. Authors need to define “good alignment” between CoT auto-evaluation and expert raters etc.

Response: We thank the reviewer for raising the point about the need to define “good alignment” for our auto-evaluation metrics. Here, we describe additional analysis to contextualize the performance of both the DDX and PACES criteria auto-evaluations described in the paper.

To quantify agreement between auto-evaluation and the specialists for diagnostic accuracy, we computed Cohen’s kappa between these two rater groups. Specifically, we utilized all predicted diagnoses from both AMIE and the PCPs and computed the Cohen’s kappa metric between the “median-specialist” ratings and auto-evaluation ratings as 0.555, indicating ‘moderate agreement’. We added the following text to appendix A.8:

“We quantify this agreement by computing Cohen’s kappa between human raters and automated evaluation with respect to the binary indicator of whether a given diagnosis, i.e., an individual item from a proposed DDX list, matched the correct final diagnosis. Cohen’s kappa for this matching task was 0.555 indicating “moderate agreement” between human raters and our automated evaluation method per established guidelines from Landis and Koch (1977).”

For the CoT auto-evaluation ratings on the PACES criteria, we used the rank-order agreement metric to determine “good alignment”. For each criteria, rank-order agreement measures the proportion of scenarios for which the evaluators agree on their preference between AMIE and the PCPs conversation (either “AMIE was better at this criteria”, “Tie”, or “PCP was better at this criteria”). 1 means the two evaluators completely agreed on their preferences, and 0 means they disagreed on all preferences. In Fig A.20, we computed the rank-order agreement between AMIE and the median specialist ratings (the red bar) as well as the average rank-order agreement between individual specialists and the median specialist rating (the blue bar). We also indicated the rank-order agreement that would be achieved with random preferences (black dashed line, always 0.33), and random preferences knowing the distribution of the median specialist’s preference (green dashed line). Our results indicate that the agreement from our CoT auto-evaluation was better than random and on par with the rank-order agreement between specialists.

Rank-order Agreements (shown in Fig A.20):

Showing Empathy:

Auto-evaluation = 0.761, Specialists = 0.730
Random = 0.33, Random with known prevalence = 0.589

Seeking and Addressing Concerns:

Auto-evaluation = 0.692, Specialists = 0.700
Random = 0.33, Random with known prevalence = 0.545

Maintaining Patient Welfare:

Auto-evaluation = 0.679, Specialists = 0.665
Random = 0.33, Random with known prevalence = 0.515

Confirming Knowledge and Understanding:

Auto-evaluation = 0.774, Specialists = 0.717
Random = 0.33, Random with known prevalence = 0.543

Comment: Section 3.4

Line 364. Does top k here mean that the diagnosis was one of the top 3 diagnoses based on the ground truth? Please define.

Response: That is correct. We thank the reviewer for raising this point of potential reader confusion, and have added the following clarification (added parts in **bold**):

*“Top-k accuracy was defined as the percentage of cases where the correct **ground truth** diagnosis appeared within the top-k positions of the DDx list. **For example, top-3 accuracy is the percentage of cases for which the correct ground truth diagnosis appeared within the top three diagnosis predictions from AMIE or PCP.**”*

Comment: Section 4

Line 378. The fact that PCPs provided fewer diagnoses (mean 5) compared to AIME (mean 10) has implications for the analyses you present in Figure 3, as the number of PCPs does not stay at 20 as one moves down the x-axis.

Response: In our Top-K Accuracy analyses, we computed for what proportion of scenarios AMIE/PCPs included the correct ground truth diagnosis (or any accepted differential) from that scenario in the first k items of their DDx. Because the minimum DDx length was 3 for PCPs, the analysis was already comparable for all $K \leq 3$. To explore whether different DDx length affects Top-K performance for the higher values of K, we recomputed the top-K accuracy with a variable K per scenario as shown in Figure R1. Specifically, for each scenario, we truncated AMIE's DDx so that its length was the same as the PCP's DDx length, and then computed the new DDx accuracy. Using this matched-length DDx, we observe that 1) Truncating the DDx had little impact on AMIE's Top-K DDx accuracy, and 2) In this matched comparison, AMIE still has significantly higher Top-K DDx accuracy for all values of K.

Figure R1: Specialist-rated top-k diagnostic accuracy with matching DDx length. AMIE and PCPs top-k DDx accuracy are compared across 159 scenarios with respect to the ground truth diagnosis (left) and all diagnoses in the accepted differential (right). Orange line represents AMIE DDx accuracy after truncating its DDx list to match the number of DDx with PCPs. Red line represents the original AMIE DDx accuracy with all 10 DDx. Bootstrapping (n=10,000) confirms all top-k differences between AMIE and PCP DDx accuracy are significant with $p < 0.05$ after FDR correction.

Comment: Section 4.2

Line 417. Explanation of non-responses by patient actors is needed, as well as how these were handled statistically.

Response: All rating items for patient actors included an option to select “Cannot rate / Does not apply”. This was by design, so patient actors were not obliged to give a response where none was applicable. For example, a Yes/No rating for “Acknowledging Mistakes” only made sense for conversations where the PCP or AMIE had in fact made a mistake spotted by the patient actor. In the absence of such a mistake, raters were asked to select “Cannot rate / Does not apply”. We explain handling of non-responses in 3.4 Statistical Analysis:

“Statistical significance for patient actor and specialist ratings was determined using two-sided Wilcoxon signed-rank tests [38] with FDR correction. Cases where either agent received “Cannot rate / Does not apply” were excluded from the test. Results below refer to p-values after FDR correction.”

Comment: Section 5

This section seems like it should be in a Discussion section (Section 6). It does not present new results but frames the work presented.

Response: We thank the reviewer for this suggestion, we will move this section (Related work) into the standalone Methods section, similar to one of our previous publications.

Comment: Section 6

Line 508. The comment about “challenging cases” suggests that AIME may be most appropriate in specialist settings, not in primary care.

Response: We thank the reviewer for pointing out potential confusion caused by this term. Here “challenging cases” refers to the NEJM challenge cases evaluated in another study [1] by our group where we found AMIE produces more accurate DDx on these challenging cases than internal medicine practitioners. We observed a similar trend in the present study with AMIE focusing on diagnostic dialogues.

[1] McDuff, D., Schaekermann, M., Tu, T., Palepu, A., Wang, A., Garrison, J., Singhal, K., Sharma, Y., Azizi, S., Kulkarni, K., 850 et al. Towards Accurate Differential Diagnosis with Large Language Models. arXiv preprint arXiv:2312.00164 (2023).

Comment: Lines 515, 518, 520. New results are presented. All results must be presented in the Results section.

Response: We thank the reviewer for the suggestion. We have moved them to the Results section.

“Accuracy by Location. We observed that both AMIE and PCPs had higher diagnostic accuracy in consultations performed in the Canada OSCE lab compared to those enacted in the India OSCE lab (see Figure A.15).

However, the differences were not statistically significant and in a subset of 40 scenarios enacted in both the Canada OSCE lab and the India OSCE lab, the performance of both AMIE and PCPs was equivalent (see Figure A.16).”

Comment: Line 553. Replace with “spent”

Response: We have incorporated the suggestion.

Comment: Line 567. This endpoint is unrealistic in the real world as well and should be acknowledged.

Response: We appreciate the suggestion and have added the following bolded text.

*“For example, in the simulated environment we impose that AMIE reaches a proposed differential and testing/treatment plan for the patient, but such an endpoint may be unrealistic for some conditions, especially in the virtual chat-based setting. **This limitation also applies in the real-world setting.**”*

Comment: Line 637. Re equity, add reference re physicians’ variation in communication skills across health literacy. See <https://www.science.org/doi/10.1126/sciadv.abj2836>

Response: We have added the suggested citation (ref. [102]).

“Other studies have found differences in physicians’ communication styles and conversation length based on gender~\cite{roter2002physician} and on patients’ level of health literacy~\cite{schillinger2021precision}.”

Comment: Line 645. Define red teaming

Response: We added the following clarification (text in bold) for red teaming.

*“Recent advances in red teaming LLMs could be useful in this scenario~\cite{perez2022red, ganguli2022red, yu2023gptfuzzer, ge2023mart}, **where human raters or other AI systems (known as the “red team”) simulate the role of an adversary to identify vulnerabilities and security gaps in these LLMs.**”*

Comment: Line 652. Choose alternate term to “low-resource languages”. Add in issues of literacy.

Response: We have changed to “minority languages”.

Comment: Limitation section needs to be expanded. Two straightforward limitations have to do with (a) toning down the issue of comprehensiveness/completeness. Few clinical presentations to primary care require a list of differential diagnoses that are 10 diagnoses long, and AMIE was programmed to generate a list of 10; and (b) expanding on the non-representativeness of patient actors and the entire experiment especially as it relates to patient’s language and literacy limitations—including their difficulties in composing and reading messages, as well as difficulties conveying clear histories and responding to clinical questions in “logical” ways (from a medical perspective).

Response:

We acknowledge the reviewer’s thoughtful comment regarding the length of AMIE’s DDX and agree that primary care typically does not involve such long differentials. As noted before, this is not necessarily a limitation and AMIE performs equally well when considering smaller differentials or matching it to the length of the PCPs (see **Figure R1** from above) and the key results of the study hold. It is quite straightforward to prompt the model to produce shorter differentials if the application or the workflow demands. We expect to conduct future studies to explore this aspect in more detail in real-world settings.

We also appreciate Referee 3’s insightful comment regarding the non-representativeness of patient actors as it relates to language and literacy.

To address this, we have taken the following steps:

- **Diversifying Dialogues:** We acknowledge that the original simulated dialogues might not fully capture the variability of real-world patient interactions. Therefore, we have enhanced our dialogue simulator to incorporate patient personality traits. Seven

distinct personalities (normal, angry and rude, chatty, sensitive, worrier, exaggerator, poor English) are now modeled, generating seven unique dialogues per vignette in the OSCE dataset. While normal patient dialogues remain concise, following instructions and directly answering questions, other personalities generate more realistic but longer dialogues, sometimes including less relevant information. This reflects the variability of real-world patient interactions.

- **Evaluating Robustness:** We have conducted additional evaluations to assess AMIE’s diagnostic capabilities when encountering these diverse personalities. As shown in the following Figure R2, AMIE demonstrates robust performance across various personality types, though minor variations in accuracy may occur under certain circumstances.

We believe these enhancements provide a more detailed understanding of AMIE’s capabilities and its ability to generalize to real-world clinical interactions. However, we note that considerable further research is needed on this aspect and consider it important future work.

Figure R2: Comparison of DDX accuracy for simulated dialogues with different patient personalities with respect to the ground truth (left) and the accepted differential (right). AMIE demonstrates robust performance across diverse personality types, with the exception of patients with “Poor English” proficiency. In these cases, patients may struggle to communicate precise and relevant information to the doctor agent, leading to lower accuracy with a relatively larger margin compared to other personality types.

Comment: But the greatest limitation is that AIME appears to have been designed and tested under an assumption that all patients have pathology. This suggests that the creators were not aware of the population-level epidemiologic realities of primary care, where well over 80% of the work in assessing novel symptoms and syndromes has to do with ruling out disease, not ruling it in. Imagine a scenario like the one the authors raise in the discussion section: one in which users (not presenting to clinical settings) use AIME like Chat GPT—engaging in diagnostic conversations with an algorithmic conversational agent whose job is to make diagnoses. The results could be disastrous—generating a population that believes it has serious pathology, the kind that demands testing and (perhaps) unnecessary utilization of services. Making a diagnosis is only a small part of clinical reasoning. This problem is exacerbated by the choice the authors

made to engage specialists (expert in pathology, who care for a small portion of the populace that indeed is ill) to assess the performance of generalists (experts in health, who care for a large portion of the populace that indeed is not ill). These conflicting roles need to be addressed and reconciled in the limitations section.

Response: We greatly appreciate the reviewer making an excellent point regarding the distribution of disease states versus non-disease states in our set of evaluation scenarios. Indeed, our original submission was based on a set of evaluation scenarios where the overwhelming majority of packs presented with a disease-state pathology as the ground truth for each scenario (see Appendix Tables A.6 to A.12), with the exception of scenarios where the ground truth was either the seeking of advice during uncomplicated pregnancy, or symptoms caused by excess caffeine intake. We agree with the reviewer that in real primary care settings, there will be patients that present with an experience of symptoms but have a non-disease state, and that it is important to be able to detect such situations. To address this important limitation, we commissioned **an additional set of ten scenarios** heavily weighted towards the ground truth being a non-disease state (as just one example, a ground truth diagnosis of “resolved constipation”), with the sole disease state being “GERD-induced chest pain” (see Appendix table A.12). We then conducted the same OSCE study flow including patient actors and specialist physician ratings for these ten extra scenarios, and conducted an ablation analysis to gauge DDX accuracy on disease states versus non-disease states. We acknowledge the nuance of having specialists perform the assessment as noted by the reviewer. Results are described in Results (Section 4.1.1) and Appendix Section A.5 in our manuscript:

***“Accuracy for Non-disease States and Positive Disease States.** Ten of the scenarios performed by AMIE and PCPs were designed to primarily describe patients without any new concerning diagnosis [e.g., a ground truth diagnosis of “resolved constipation”, and one recurrence of a prior known disease-state of gastroesophageal reflux disease (GERD) induced chest pain]. These were two scenarios each from Cardiovascular, Gastroenterology, Internal Medicine, Neurology, and Respiratory specialties. Here, we plot the top-k DDX accuracy as rated by the majority vote of three specialists for these “non-disease state” cases. Though our results are not statistically significant, as they only consist of 10 scenarios, AMIE appears to maintain the same trend of better performance on these mostly negative scenarios. AMIE also has superior DDX accuracy on the set of 149 primarily positive disease state scenarios (in which only 3 scenarios had a ground truth of a non-disease state).”*

Figure A.7 | Specialist rated DDX accuracy on non-disease state patients. (a) Specialist rated DDX top-k accuracy for the 10 “negative” scenarios conducted by AMIE with respect to the ground truth diagnosis. (b) Specialist rated DDX top-k accuracy for the 10 “negative” scenarios conducted by AMIE with respect to the ground truth diagnosis. None of the differences between AMIE and PCP DDX accuracy are statistically significant (bootstrap with $n=10,000$ and FDR correction) due to the low number of scenarios being tested.

Figure A.8 | Specialist rated DDX accuracy on patients with pathology. (a) Specialist rated DDX top-k accuracy for the 149 “positive” scenarios conducted by AMIE with respect to the ground truth diagnosis (146/149 were disease-positive states). (b) Specialist rated DDX top-k accuracy for the 149 “positive” scenarios conducted by AMIE with respect to the ground truth diagnosis (146/149 were disease-positive states). All differences between AMIE and PCP DDX accuracy are statistically significant (bootstrap with $n=10,000$ and FDR correction).

All figures throughout the paper were updated to reflect the increased set of **159** scenarios (149 with 146/149 positive disease states + 10 with non-disease states (1/10 being a recurrence of a known disease state)).

While we believe that this additional data increases confidence in AMIE’s ability to distinguish pathology from non-disease states, we acknowledge this set of scenarios is limited in size and therefore does not permit conclusive insights. We expanded our Discussion section (Breadth of Evaluation) as follows:

“Note that the majority of scenarios in our evaluation set assume an underlying disease state while only a small subset assumes the absence of disease. This is an important limitation of this work as it does not reflect the population-level epidemiologic realities of primary care, where the majority of work in assessing patients involves ruling out disease, rather than ruling it in. We encourage future work to explore evaluation with various distributions of disease vs. non-disease states.”

Comment: Many Figures (A.9, A15) may need to be presented as Supplemental Tables or information. Right now, much is not formatted for a scientific journal such as Nature. The journal editors will need to make specific recommendations based on their standards.

Response: We thank the reviewer for the suggestions, we will reformat the entire manuscript to conform with Nature guidelines following the editorial team’s recommendations upon the acceptance for publication.

From Nature Formatting Guidelines:

“Contributions should be organized in the sequence: title, authors, affiliations (plus present addresses), bold first paragraph, main text, main references, tables, figure legends, methods (including separate data and code availability statements), methods references, acknowledgements, author contributions, competing interest declaration, additional information (containing supplementary information line (if any) and corresponding author line), extended data figure/table legends.”

Comment: Code was not provided.

Response: We acknowledge this limitation. Our goal is to innovate responsibly. We believe that systems such as those described in our work hold great promise but the safety-critical nature of the medical domain requires careful considerations and oversight (including equity, policy, regulatory and liability considerations). We have attempted to describe our methods in detail so that interested researchers can build on our work using other LLMs available to them. We also note the availability of some of the models used in the work (PaLM 2 and Med-PaLM 2) via Google Cloud APIs as well as open source versions of related performant models (<https://ai.google.dev/gemma>).

We once again thank the reviewers and editors for the consideration of our manuscript.

Referees' comments:

Referee #2 (Remarks to the Author):

Overall, I am impressed that the authors have responded very well to both my critique of the original submission and the comments from Reviewer #3. I had felt that the work reported here is quite innovative, important, and carefully done, so it deserves a high profile publication. The additions spurred by our critiques make it even better.

Most of my residual issues in this second review are just editing questions:

Fig 1: overlap of “Outer Self-play” with “Patient’s Confidence”

We have fixed it.

356-360 repeats description of the randomization of order in which a patient actor interacts with a PCP or AMIE.

We changed to:

“The ordering of PCP and AMIE was randomized and patient actors were not informed as to which they were talking to in each consultation (counterbalanced design to control for any potential order effects).”

Figs A.7 and A.8: The captions duplicate the description of the (a) graphs and (b) graphs, whereas obviously the two report different results.

Thank you for catching this error in our captions for figs A.7 and A.8, now put together in the Extended Data (Figure ED.2). These are indeed not the same; we have updated the caption accordingly for the subplots, which display accuracy with respect to ground truth (a, c) vs. any accepted differential (b, d).

It occurred to me that the authors seem to have missed an opportunity, in the spirit of Turing, to ask patient actors to identify whether they thought they were conversing with a PCP or AMIE. The result might shed further light on their discussion of the artificiality of the communication scenario and the observed differences between judgments of PCPs and AMIE.

--Peter Szolovits

Thanks so much for the excellent feedback and constructive reviews. The goal of our study was trying to understand the efficacy of the AMIE system in performing diagnostic consultations in simulated settings but not necessarily to exactly mimic and match a human doctor in their behavior. As such a Turing study endpoint felt out of scope for the work. While we did our best to control and randomize the study and ensure its double blind design (for example, by adding delays to the AI responses), we expect that if directly asked patient actors often would have been able to distinguish between the AI and human doctors based on the nature of the responses and interaction.

Referee #3 (Remarks to the Author):

The authors did a remarkable job addressing this reviewer's concerns and I believe that the revised manuscript should be accepted under two contingencies:

1. The exploratory findings re the poor performance of AIME under the scenario in which the patient has limited English skills should be included in this paper (not just a response to a reviewer), as it has terribly important relevance to the earlier discussion re health equity. Including these results does not take away from the transformative nature of this work; rather, it points to the intent of the team to continue to refine the tool to ensure that its benefits do not only accrue to those with the least needs for such innovation. If desired, the authors could limit the results to just those with limited English proficiency, as opposed to the other "personality types."

Thanks so much for the constructive feedback. We have included these exploratory findings for the 7 "personality types" as supplementary information in the manuscript (Figure SI.3).

We reference this in the main text results (Section 2.1.2 Efficiency in acquiring information):
"Additionally, we present a fully-simulated ablation testing different patient behaviors (see Figure SI.3), which found that AMIE was robust to many different patient personalities, though it struggled when interviewing patients with low English literacy."

We also added the following in the discussion: *"Though the simulated patients encompassed a wide range of conditions, they failed to capture the full range of potential patient backgrounds, personalities, and motivations. Indeed, our simulated experiments in Figure SI.3 suggested that while AMIE appears robust to certain variations in patient characteristics and behavior, it has significant difficulty with some types of patients, such as those with low English literacy."*

2. The necessary changes in formatting and placement of sections will need to be carefully reviewed by the editor, as for the most part, the authors did not address these at this point in their revise and resubmit.

Thanks so much, we have gone through the manuscript and made changes based on the editorial guidelines. We look forward to working with the editorial team to make any additional necessary changes to fit the Nature publication formats and standards.

Referee #3 (Remarks on code availability):

As I had mentioned when i was invited to review this work, I am not expert in LLM coding.